# Towards Unified Multimodal Editing
# with Enhanced Knowledge Collaboration

**Kaihang Pan**[1,*], **Zhaoyu Fan**[1,*], **Juncheng Li**[1,†], **Qifan Yu**[1], **Hao Fei**[2]
**Siliang Tang**[1], **Richang Hong**[3], **Hanwang Zhang**[4], **Qianru Sun**[5]

Zhejiang University[1], National University of Singapore[2]
Hefei University of Technology[3], Nanyang Technological University[4]
Singapore Management University[5]
{kaihangpan, zyfan, junchengli, yuqifan, siliang}@zju.edu.cn
haofei37@nus.edu.sg, hongrc.hfut@gmail.com
hanwangzhang@ntu.edu.sg, qianrusun@smu.edu.sg

## Abstract

The swift advancement in Multimodal LLMs (MLLMs) also presents significant challenges for effective knowledge editing. Current methods, including intrinsic knowledge editing and external knowledge resorting, each possess strengths and weaknesses, struggling to balance the desired properties of reliability, generality, and locality when applied to MLLMs. In this paper, we propose **UniKE**, a novel multimodal editing method that establishes a unified perspective and paradigm for intrinsic knowledge editing and external knowledge resorting. Both types of knowledge are conceptualized as vectorized key-value memories, with the corresponding editing processes resembling the assimilation and accommodation phases of human cognition, conducted at the same semantic levels. Within such a unified framework, we further promote knowledge collaboration by disentangling the knowledge representations into the semantic and truthfulness spaces. Extensive experiments validate the effectiveness of our method, which ensures that the post-edit MLLM simultaneously maintains excellent reliability, generality, and locality. The code for UniKE is available at https://github.com/beepkh/UniKE.

## 1 Introduction

The rapid development of Large Language Models (LLMs) [35, 36] has made it increasingly important to ensure the real-time accuracy of their outputs in an efficient manner. To this end, in the NLP community, **Knowledge Editing** [41, 45] has been proposed as a data- and time-efficient way to edit LLMs, correcting errors or outdated responses while ensuring no negative impacts are created. The post-edit model is required to generate the desired output given the input (**Reliability**), also generalize over other equivalent neighbors of inputs (**Generality**) without altering the output over other irrelevant inputs (**Locality**). Knowledge editing methods can be divided into two main categories based on the type of knowledge involved: **intrinsic knowledge editing** [11, 28] where we update specific model parameters to store new knowledge in a parametric manner; **external knowledge resorting** [46, 29] that LLMs perceive the new knowledge contained in the relevant context (e.g., via in-context learning). Both types of methods have shown good effectiveness in editing LLMs.

Going a step further, with the emergence of advanced multimodal large language models (MLLMs [1]), there has been a further exploration into **Multimodal Editing**. Unfortunately, [4] finds that though

---

[*] Equal Contribution.
[†] Corresponding Author.

38th Conference on Neural Information Processing Systems (NeurIPS 2024).

| Method | Knowledge Type | Knowledge Form | Locality | Generality |
|---|---|---|---|---|
| Intrinsic Knowledge Editing | Intrinsic Knowledge | Parametric Neurons | 👎 | 👎 |
| External Knowledge Resorting | External (In-context) Knowledge | Descriptive Examples | 👎 | 👍 |
| UniKE | Intrinsic & In-context Knowledge | Unified Vectorized Key-Value Pairs | 👍 | 👍 |

Figure 1: Comparisons of existing knowledge editing methods and UniKE.

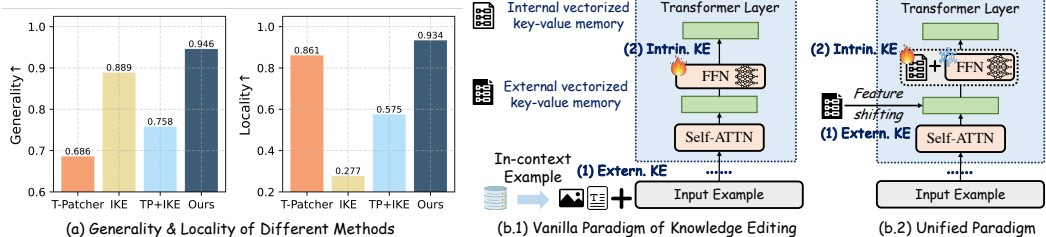

Figure 2: **(a)** The generality and locality on MMEdit [4] when applying T-Patcher [11] (intrinsic knowledge editing), IKE [46] (external knowledge resorting), the combination of these two (TP+IKE), and UniKE for multimodal editing. **(b)** The paradigm of intrinsic knowledge editing (Intrin. KE) and external knowledge resorting (Extern. KE) before and after knowledge unification.

efficient in editing LLMs, existing methodologies face considerable challenges for MLLMs due to the inherent diversity and complexity of multimodal knowledge. Despite still maintaining high reliability, **they struggle to simultaneously achieve both ideal locality and generality**, as shown in Figure 1.

We argue that both approaches, whether intrinsic knowledge editing or external knowledge resorting, have respective drawbacks for multimodal editing. Specifically, **intrinsic knowledge editing** (*e.g.*, T-Patcher [11] that integrates additional neurons into MLLM) tries to eliminate the risk of losing previously-learned facts and preserve locality. However, it also leads to the newly integrated knowledge resembling rote memorization [3] with **weak generality of its truthfulness**, as multimodal reasoning requires a coordinated understanding of semantics from multiple modalities. Conversely, though **external knowledge resorting** (*e.g.*, in-context editing [46]) retrieves generalizable information from external databases, the in-context knowledge **may not have a strong semantic relevance** with the original input [23]. This can mislead the MLLM into areas they originally excelled, resulting in **weak locality**. Figure 2.a provides direct evidence to support the above discussion.

Therefore, how can we effectively edit MLLMs? One intuitive idea lies in directly combining intrinsic knowledge editing with external knowledge resorting, leveraging the advantages of both. However, in intrinsic knowledge editing (such as T-Patcher), the extra integrated knowledge typically incorporates parametric neurons into the model parameters, which is abstract with high-level semantics. Conversely, external knowledge resorting, such as in-context editing, feeds the MLLM with descriptive images and text at the input end, directly describing the content with low-level semantics. Consequently, these two methods exhibit **significant differences in paradigms at inconsistent semantic levels** and it is **challenging to establish a synergistic correlation** with each other. Figure 2.a demonstrates that simply combining T-Patcher and in-context editing leads to both undesirable locality and generality in the post-edit MLLM, highlighting the drawbacks of each approach separately.

To address the above issue, we propose **UniKE**, a novel multimodal editing method that establishes a unified framework for both intrinsic knowledge editing and external knowledge resorting, enabling a synergistic knowledge collaboration. First, we develop a unified view for intrinsic and external knowledge, both represented as **vectorized key-value memories** at the same semantic levels. Based on this view, we combine both types of knowledge editing methods, executing them in the latent space with a unified paradigm, as shown in Figure 2.b. Specifically, intrinsic knowledge editing integrates extra knowledge into the internal key-value memory at the feed-forward network; external knowledge resorting leverages an external key-value memory to inject knowledge into self-attention via feature shifting. Both methods could be performed in the same transformer layers with a synergistic correlation, preliminarily allowing each to utilize strengths for complementing the other.

Moreover, we further effectively enhance the collaboration between intrinsic knowledge and external knowledge resorting. Within the unified framework, the two editing methods still require emphasis on different aspects of knowledge to further complement their respective drawbacks: intrinsic knowledge should focus on generalizable **truthfulness**, while external knowledge should have relevant **semantics**

to the input samples. So we leverage contrastive learning to disentangle the knowledge representations into the semantic and truthfulness spaces. In the semantic space, we enable the intrinsic knowledge to assist in selecting appropriate external knowledge with its inclusion magnitude, ***preventing the disruption of locality***. Simultaneously, in the truthfulness space, we employ the external knowledge to identify a generalizable editing direction to regulate the integrated intrinsic knowledge, ***alleviating its restriction of generality.*** Under such a synergistic promotion, extensive experiments show that UniKE achieves promising results under various settings, ensuring that the post-edit MLLM maintains excellent reliability, generality, and locality. Overall, our main contributions are three-fold:

- We propose a unified paradigm for multimodal knowledge editing, with both intrinsic and external knowledge represented as vectorized key-value memories, conducting at the same semantic levels in the same transformer layers.
- We disentangle the knowledge representations into the semantic and truthfulness spaces, promoting the collaboration between intrinsic knowledge editing and external knowledge resorting.
- Our method ensures that, under various backbones and editing scenarios, the post-edit MLLM consistently possesses all three properties well.

## 2  Related Work

Recent years witness a burgeoning in the techniques of knowledge editing for LLMs [41, 45], with the post-edit model expected to exhibit three properties [11]: Reliability, Generality, and Locality (Detailed definitions are given in Appendix B). Knowledge editing methods can be divided into two main categories based on the type of knowledge: intrinsic knowledge editing and external knowledge resorting. Intrinsic knowledge editing [6, 28, 25, 26], involves the parametric storage of knowledge within the model, requiring modifications to LLMs' parameters. While external knowledge resorting [46, 29, 47] typically preserves LLMs' parameters and maintains a knowledge database to retrieve relevant cases for each input with several information retrieval approaches [13, 31]. Overall, intrinsic and external knowledge exhibit significant differences in the knowledge forms (parametric neurons and descriptive in-context examples, respectively).

Furthermore, the emergence of MLLMs [1, 48, 19, 33, 40, 8] has sparked several studies on multimodal knowledge editing [4, 30]. However, [4] find that existing methods fall short of expectations when editing MLLMs. Though maintaining high reliability, whether intrinsic knowledge editing or external knowledge resorting, often fails to simultaneously achieve ideal locality and generality as shown in Figure 1. In this paper, we propose a synthesis of both types of methods for multi-modal editing. By unifying intrinsic and in-context knowledge as vectorized key-value memories, we facilitate collaborative interaction between the two within the unified paradigm, fully utilizing the strengths of each method and enabling the post-edit MLLM to consistently exhibit all three properties well.

## 3  Method

In this section, we first develop a unified view for knowledge editing (§3.1). Within the unified framework, we introduce how to realize intrinsic knowledge editing and external knowledge resorting in the latent space (§3.2). Finally, we further enhance the collaboration between both types of knowledge to control the overall editing process (§3.3). The overall framework is shown in Figure 3.

### 3.1  A Unified View for Knowledge Editing

In our general understanding, intrinsic knowledge editing and external (in-context) knowledge resorting seem to have stark differences. In this section, we will demonstrate that both intrinsic and in-context knowledge can be unified as **vectorized key-value memories**, directly acting on the hidden states within the transformer. Consequently, knowledge editing can be understood as adjusting the key-value pairs in the memory to activate appropriate knowledge for a given query representation.

**Intrinsic Knowledge as Internal Key-Value Memory.**   Previous studies have demonstrated that the feed-forward network (FFN) in the transformer harbors a wealth of knowledge [5, 25]. We aim to conduct intrinsic knowledge editing within the FFN and treat FFN as parametric key-value memory

Figure 3: (a) We develop a unified view for multimodal editing, with both intrinsic and external knowledge represented as vectorized key-value memory. (b) We disentangle the knowledge representation into the semantic and truthfulness spaces, further enhancing the knowledge collaboration.

storing within the MLLM. Considering a two-layer FFN: given the input latent states, the FFN treats it as a query $q$, with the first FFN layer acting as keys $W_K^{ffn} \in \mathbb{R}^{d \times d'}$, and the second layer as values $W_V^{ffn} \in \mathbb{R}^{d' \times d}$. Here, $d$ is the hidden dimension of the FFN; and $d'$ is the intermediate hidden dimension of the FFN, also interpreted as the memory size. Consequently, FFN effectively uses the query to match the keys, with the intermediate hidden state $o$ representing the weight for each value in memory. FFN then outputs the weighted sum of all values $\text{FFN}(q)$.

$$o = \text{Act}(qW_K^{ffn} + b_K^{ffn}) \quad \text{\# Matching keys with query.}$$
$$\text{FFN}(q) = oW_V^{ffn} + b_V^{ffn} \qquad \text{\# Outputting the weighted sum of values.}$$

(1)

where $b_K^{ffn} \in \mathbb{R}^{d'}$, and $b_V^{ffn} \in \mathbb{R}^{d}$ are two bias vectors. $Act(\cdot)$ is a non-linear activation function.

**In-context Knowledge as External Key-Value Memory.** Traditional in-context knowledge increases the context window space and makes it difficult to quantitatively control [27]. Here we propose a similar view of in-context knowledge as an external vectorized key-value memory to establish its connection with intrinsic knowledge. Specifically, in-context learning typically concatenates the external multimodal knowledge $X_{know}$ with the original input sequence $X_{input}$ to form the combined sequence $X = [X_{know}, X_{input}]$. Considering the self-attention mechanism $\text{Attn}(Q = X, K = X, V = X)$ in the transformer, during in-context learning, the attention $\text{Attn}(X_{input}, X, X)$ for tokens in the original input sequence can actually be formulated as follows:

$$\text{Attn}(X_{input}, X, X) = \alpha \underbrace{\text{Attn}(X_{input}, X_{input}, X_{input})}_{\text{h}_{input}} + (1 - \alpha) \underbrace{\text{Attn}(X_{input}, X_{know}, X_{know})}_{\text{h}_{know}}$$

(2)

The first term $\text{h}_{input}$ is the original self-attention output without in-context knowledge. The second term is to treat the hidden states of $X_{know}$ as a key-value memory, the hidden state of $X_{input}$ as the query, selectively activating the relevant in-context knowledge $\text{h}_{know}$. $\text{h}_{know}$ then performs position-wise feature shifting on the original attention output to achieve in-context editing, with $\alpha$ as the scaling factor. We give a more complete analysis in Appendix A.

### 3.2 Unified Knowledge Editing within Latent Space

**Assimilation: Intrinsic Knowledge Editing.** As intrinsic knowledge is considered as key-value memory stored within the FFN, akin to [11], we treat intrinsic knowledge editing as the process of integrating extra knowledge into the internal knowledge memory, thereby establishing connections with prior knowledge. This process is analogous to the **Assimilation** phase [34] in human cognition, where an individual incorporates new knowledge into their existing cognitive structures. Specifically, based on the analysis in Eq.(1), the newly added parametric knowledge is stored in the FFN as key-value pairs (the number of new pairs is $ne$), transforming the output of the FFN as:

$$\begin{bmatrix} o & o_{extra} \end{bmatrix} = \text{Act}(q \begin{bmatrix} W_K^{ffn} & W_K^{extra} \end{bmatrix} + \begin{bmatrix} b_K^{ffn} & b_K^{extra} \end{bmatrix})$$
$$\text{FFN}_{edit}(q) = \begin{bmatrix} o & o_{extra} \end{bmatrix} \cdot \begin{bmatrix} W_V^{ffn} \\ W_V^{extra} \end{bmatrix} + b_V^{ffn} = \text{FFN}(q) + o_{extra}W_V^{extra}$$

(3)

where $W_K^{extra} \in \mathbb{R}^{d \times ne}$ and $W_V^{extra} \in \mathbb{R}^{ne \times d}$ are the extra keys and values, $b_K^{extra} \in \mathbb{R}^{ne}$ is an extra bias vector. $o_{extra} = \text{Act}(q \cdot W_K^{extra} + b_K^{extra})$ represents the activated weight of the extra value. In this way, the newly injected knowledge, in the form of key-value pairs, is seamlessly integrated into the existing knowledge structure of MLLM.

**Accommodation: External Knowledge Resorting.**   As in-context knowledge can also be vectorized into hidden states as external key-value memory, we can interpret in-context knowledge editing as shifting the original post-attention latent states as shown in Eq.(2), which also allows in-context knowledge to be introduced in a more controlled manner. This process is analogous to the **Accommodation** phase in human cognition, where individuals modify their existing cognitive schemas to accommodate new information that does not fit into their prior understanding.

Assuming the hidden states of in-context knowledge have been extracted and stored as key-value pairs $\mathcal{M}_{ext} = \{(h_{sem}, h_{pos})\}$, in the input end, the MLLM is fed only the original sample without concentrating in-context knowledge. Within a given transformer layer, we initially utilize the pre-attention hidden states $\text{h}_{input}^{pre}$ to retrieve the top-K $\{h_{sem,i}\}_{i=1}^{K}$ that exhibit the highest cosine similarity from $\mathcal{M}_{ext}$, obtaining the corresponding $\{h_{pos,i}\}_{i=1}^{K}$. As indicated in Eq.(2), $\{h_{pos,i}\}_{i=1}^{K}$ then serves as both keys and values for attention computation, with $h_{input}^{pre}$ acting as the query, thereby achieving the in-context latent states $\text{h}_{know}$. Subsequently, by simply specifying a scalar $\alpha$, $\text{h}_{know}$ is integrated with the original self-attention output $\text{h}_{input}$, acting as a shifting direction that steers the original states closer to the representations of in-context knowledge, thus facilitating editing.

**Analysis of the Unified Framework.**   In real life, assimilation and accommodation work together with ongoing interaction to drive cognitive development. Within the unified knowledge editing framework, we also inherently establish a **preliminary collaboration** between intrinsic knowledge editing and external knowledge resorting: *external knowledge assists in storing more generalizable intrinsic knowledge; intrinsic knowledge helps to select appropriate external knowledge.* As shown in Figure 3.a, in the $l$-th transformer layer, the post-self-attention states following in-context editing, are directly fed into the FFN for intrinsic knowledge editing. when the FFN input integrates generalizable in-context knowledge, the newly added key-value pairs in FFN also tend to store generalizable knowledge to be better activated. Moreover, the output of the FFN, having just undergone intrinsic knowledge editing, is transmitted to the self-attention of the $(l + 1)$ layer. Here, it acts as the query to select suitable hidden states of in-context knowledge for in-context editing. Overall, compared to directly combining different knowledge editing methods with various paradigms, we establish *a synergistic correlation with the unification of knowledge editing paradigm, allowing different methods to utilize their strengths to complement each other.*

### 3.3   Enhanced Collaboration with Knowledge Disentangling

To further promote the collaboration between intrinsic knowledge editing and external knowledge resorting, it is essential to emphasize different aspects of knowledge: intrinsic knowledge should prioritize generalizable **truthfulness** to improve generality, whereas external knowledge should maintain **semantic** relevance to the input samples to preserve locality. Inspired by this, we extract diverse hidden states for in-context knowledge and innovatively disentangle the knowledge representations into semantic and truthfulness spaces, further enhancing the collaboration within these two spaces.

**Extracting In-context Knowledge Representations.**   To construct the representations of in-context knowledge, we first acquire knowledge that the MLLM has not previously mastered, and collect triplets $\{(Q_I, A_{pos}, A_{neg})\}$. $Q_I$ is the input multimodal question, $A_{pos}$ is the truthful answer, $A_{neg}$ is the MLLM's hallucinated prediction. For each piece of knowledge, we pair $Q_I + A_{pos}$ as the positive knowledge, $Q_I + A_{neg}$ as the negative knowledge, and separately pass the positive and negative knowledge through the MLLM, obtaining three critical hidden states. **Semantic hidden state** $h_{sem}$ is related to the last token of the question part before MLLM processes the response, inherently encoding the semantic information on the given examples. **Positive hidden state** $h_{pos}$ and **negative hidden state** $h_{neg}$ correspond to the final token of the entire input from the positive and negative knowledge, respectively. They provide insights into how the responses guide the MLLM onto the correct or incorrect track. Note that we store $(h_{sem}, h_{pos})$ as the key-value pairs in the knowledge memory for in-context editing in §3.2. More details are given in Appendix C.

**Disentangling Knowledge Representations.** Then we explicitly disentangle the representations of in-context knowledge into semantic and truthfulness spaces. Within the semantic space, $h_{pos}$ and $h_{neg}$ (along with the semantic hidden states $h_{sem}$) from the same sample encapsulate identical meanings; whereas in the truthfulness space, $h_{pos}$ and $h_{neg}$ must be distinctly differentiated.

Specifically, we introduce a truthfulness encoder $\mathtt{Enc}^{Tru}(\cdot)$ and a semantic encoder $\mathtt{Enc}^{Sem}(\cdot)$, mapping each pair of $\{h_{pos}, h_{neg}\}$ to the semantic and truthfulness space, deriving a set of semantic representations $(H_{pos}^{Sem}, H_{neg}^{Sem})$ and truthfulness representations $(H_{pos}^{Tru}, H_{neg}^{Tru})$, respectively. Within these two latent spaces, we leverage contrastive learning to probe representations with similar truthfulness but different semantics, and conversely, those that are semantically similar but differ in truthfulness. In the **truthfulness space**, for the given positive or negative truthfulness representations $h^{Tru} = h_{pos,i}^{Tru} \ (h_{neg,i}^{Tru})$, examples sharing the same truthfulness $H_{pos}^{Tru} \ (H_{neg}^{Tru})$ form $S^+$, while those with opposite truthfulness $H_{neg}^{Tru} \ (H_{pos}^{Tru})$ form $S^-$. The objective of contrastive learning is to minimize the distance between $h^{Tru}$ and $S^+$ while maximizing the distance between $h^{Tru}$ and $S^-$:

$$\mathcal{L}_1(h^{Tru}, S^+, S^-) = \sum_{i=1}^{n} \left( -log \frac{\sum_{h \in H_{pos}^{Tru}} exp(s(h_{pos,i}^{Tru}, h))}{\sum_{h \in (H_{pos}^{Tru}, H_{neg}^{Tru})} exp(s(h_{pos,i}^{Tru}, h))} - log \frac{\sum_{h \in H_{neg}^{Tru}} exp(s(h_{neg,i}^{Tru}, h))}{\sum_{h \in (H_{pos}^{Tru}, H_{neg}^{Tru})} exp(s(h_{neg,i}^{Tru}, h))} \right)$$
(4)

where $s$ is the similarity function. In the **semantic space**, for a given semantic hidden state $h_{sem,i}$, its corresponding semantic representations ($h_{pos,i}^{Sem}$ and $h_{neg,i}^{Sem}$) form the $S^+$, while those from other examples $H_{pos}^{Sem} \backslash h_{pos,i}^{Sem}, H_{neg}^{Sem} \backslash h_{neg,i}^{Sem}$ form $S^-$. And the loss of contrastive learning is:

$$\mathcal{L}_2(h_{sem}, S^+, S^-) = \sum_{i=1}^{n} -log \frac{exp(s(h_{sem,i}, h_{pos,i}^{Sem})) + exp(s(h_{sem,i}, h_{neg,i}^{Sem}))}{\sum_{h \in H_{pos}^{Sem}} exp(s(h_{sem,i}, h)) + \sum_{h \in H_{neg}^{Sem}} exp(s(h_{sem,i}, h))}$$
(5)

**Enhanced Knowledge Collaboration within Disentangled Spaces.** After knowledge disentangling, we could further enhance the knowledge collaboration within the two spaces. Specifically, In the **truthfulness space**, we calculate the average truthfulness representations ($\hat{H}_{pos}^{Tru}$ and $\hat{H}_{neg}^{Tru}$) over all positive and negative hidden states of in-context knowledge, to regulate intrinsic knowledge editing. As the representations of positive and negative hidden states exhibit distinct truthfulness after training, we identify an editing direction $\zeta = \hat{H}_{pos}^{Tru} - \hat{H}_{neg}^{Tru}$, pointing from the center of untruthful representations to the center of truthful representations. And then we utilize a learnable weight $W_\zeta$ to map $\zeta$ from the truthfulness space back to the representation space: $\zeta' = W_\zeta \zeta$. On this basis, during intrinsic knowledge editing in Eq.(3), we further combine $W_V^{extra}$ with $\zeta'$ as follows:

$$\mathtt{FFN}_{edit}(q) = \begin{bmatrix} o & o_{extra} \end{bmatrix} \cdot \begin{bmatrix} W_V^{ffn} \\ W_V^{extra} + \beta \cdot \zeta' \end{bmatrix} + b_V^{ffn} = \mathtt{FFN}(q) + o_{extra}(W_V^{extra} + \beta \cdot \zeta')$$
(6)

where $\beta$ is an editing scalar. In the **semantic space**, as we leverage $\alpha$ in Eq.(2) to control the inclusion magnitude of in-context knowledge, we further leverage the hidden states after intrinsic knowledge editing to adaptively control $\alpha$. Based on $\mathtt{h}_{know}, \mathtt{h}_{input}$ in Eq.(2), we first extract the semantic representations of the injected in-context knowledge $h_{know}^{sem} = \mathtt{Enc}^{Sem}(\mathtt{h}_{know})$ and the hidden states from the last token of $\mathtt{h}_{input}$ ($\mathtt{h}_{input}[-1]$ serves a similar role as the semantic hidden state $h_{sem}$). We then assign the cosine similarity between $h_{know}^{sem}$ and $\mathtt{h}_{input}[-1]$ to $\alpha$ , with Eq.(2) reformulated as:

$$\mathtt{Attn}(X_{input}, X, X) = \mathtt{Sim}(h_{know}^{Sem}, \mathtt{h}_{input}[-1]) \cdot \mathtt{h}_{input} + \left(1 - \mathtt{Sim}(h_{know}^{Sem}, \mathtt{h}_{input}[-1])\right) \cdot \mathtt{h}_{know}, \quad (7)$$

**Analysis of Knowledge Collaboration.** In the **truthfulness space**, $\zeta$ is derived from the distribution deviation between hallucinated knowledge and truthful knowledge based on a large number of examples. As the newly integrated intrinsic knowledge is typically learned from a single editing sample which easily leads to overfitting, $\zeta$ effectively regulates the values of new intrinsic knowledge into a generalizable truthful direction to improve generality. **In the semantic space**, when the relevance between the in-context knowledge and the input sample is weak, $\alpha = \mathtt{Sim}(h_{know}^{sem}, \mathtt{h}_{input}[-1])$ adaptively takes a small value thanks to contrastive training. As external knowledge resorting needs to prevent the excessive inclusion of unrelated external knowledge, a smaller $\alpha$ effectively reduces its inclusion magnitude to preserve locality. We further provide quantitative analysis in §4.5.

Table 1: Main results on one-step editing on the MMEdit. Rel., T-Gen., M-Gen., T-Loc., and M-Loc. refer to Reliability, T-Generality, M-Generality, T-Locality, and M-Locality, respectively.

| Method | EDITING VQA (E-VQA) | | | | | EDITING IMAGE CAPTION (E-IC) | | | | |
|---|---|---|---|---|---|---|---|---|---|---|
| | Rel. ↑ | T-Gen. ↑ | M-Gen. ↑ | T-Loc. ↑ | M-Loc. ↑ | Rel. ↑ | T-Gen. ↑ | M-Gen. ↑ | T-Loc. ↑ | M-Loc. ↑ |
| **BLIP-2 OPT** | | | | | | | | | | Size: 3.8B |
| Backbone Model | 0.0 | 0.0 | 0.0 | 100.0 | 100.0 | 0.0 | 0.0 | 0.0 | 100.0 | 100.0 |
| FT (last layer) | 58.7 | 54.2 | 49.4 | 67.7 | 63.1 | 61.1 | 52.1 | 51.6 | 55.0 | 49.5 |
| KE | 85.3 | 77.4 | 75.3 | 93.8 | 66.4 | 50.5 | 49.0 | 46.3 | 95.0 | 64.3 |
| T-Patcher | 85.6 | 80.3 | 74.6 | 90.5 | 89.7 | 85.6 | 73.4 | 70.0 | 91.1 | 82.0 |
| MEND | 99.4 | 98.8 | 79.1 | **99.9** | 96.6 | 96.1 | 95.8 | 74.2 | 94.5 | 70.8 |
| In-Context Editing | **99.7** | 93.9 | 93.6 | 48.8 | 2.5 | 96.7 | 78.2 | 87.6 | 49.0 | 3.0 |
| SERAC | 99.4 | **99.4** | 86.8 | 96.8 | 2.9 | **99.7** | **98.9** | 89.2 | 95.7 | 7.5 |
| **UniKE (Ours)** | 98.8 | 98.4 | **94.8** | 98.3 | **96.7** | 98.3 | 96.3 | **93.2** | 95.8 | **85.7** |
| **MiniGPT-4** | | | | | | | | | | Size: 7.3B |
| Base Model | 0.0 | 0.0 | 0.0 | 100.0 | 100.0 | 0.0 | 0.0 | 0.0 | 100.0 | 100.0 |
| FT (last layer) | 70.1 | 65.7 | 63.9 | 72.6 | 65.8 | 67.4 | 65.1 | 62.8 | 63.5 | 52.7 |
| KE | 91.8 | 89.0 | 60.8 | 96.9 | 67.8 | 96.6 | 67.8 | 57.4 | 97.3 | 64.4 |
| T-Patcher | 83.0 | 68.2 | 66.0 | 84.8 | 82.0 | 83.8 | 72.3 | 67.7 | 93.9 | 83.6 |
| MEND | 98.8 | **98.6** | 82.2 | 98.2 | 81.1 | 96.6 | **96.1** | 76.3 | 98.4 | 75.3 |
| In-Context Editing | **100.0** | 94.9 | 90.5 | 50.3 | 3.7 | 90.9 | 81.6 | 88.5 | 52.2 | 4.7 |
| SERAC | 87.7 | 87.6 | 85.9 | 97.5 | 14.2 | 91.8 | 91.4 | 91.0 | 97.9 | 7.2 |
| **UniKE (Ours)** | 98.0 | 97.4 | **92.8** | **98.7** | **88.8** | **96.8** | 95.7 | **92.4** | **98.9** | **87.3** |

# 4 Experiments

We first evaluate UniKE on **one-step editing** (§4.2), the standard setup of multimodal editing. We further extend the setup to **sequential editing** (§4.3) and **cross-task editing** (§4.4) for evaluation.

## 4.1 Experimental Setup

**Dataset & Backbone & baselines.** Our experiments are conducted on the MMEdit benchmark [4], which contains two subtasks: Editing VQA (E-VQA) and Editing Image Caption (E-IC). We leverage Reliability, generality (T-Generality and M-Generality) and locality (T-Locality and M-Locality) as the evaluation metrics. For one-step editing, we conduct experiments on BLIP2-OPT [19] and MiniGPT-4 [48]; for sequential editing and cross-task editing, we conduct experiments on MiniGPT-4.

Furthermore, We use the following baselines: **(1) Fine-tuning method:** tuning the last layer of MLLM; **(2) Intrinsic knowledge editing method:** Knowledge Editor (KE) [5], MEND [28], T-Patcher [11]; **(3) External knowledge resorting method:** In-Context Editing (IKE) [46], SERAC [29].

**Implementation Details.** We conduct knowledge editing in the latent space. In intrinsic knowledge editing, we add extra key-value pairs into the last four transformer layers; In external knowledge resorting, we retrieve top-40 in-context hidden states for each case and conduct feature shifting in the last four layers. More details of the experimental setup are shown in Appendix D.

## 4.2 Main Results on One-step Editing

Table 1 shows the results of one-step editing, where each edit aims to correct a single mistake. We further provide a statistical summary in Append D.4. We have the following observations: *(i)* Most knowledge editing methods could achieve desirable reliability. *(ii)* Despite achieving high locality, **most intrinsic knowledge editing methods have room for improvement in generality** (*e.g.*, average generality and locality of T-Patcher across all settings are 71.6 and 87.2). *(iii)* Although achieving commendable generality, **the locality of external knowledge resorting methods is not ideal**. Specifically, the average locality (generality) of IKE and SERAC are 26.8 (88.6) and 52.5(91.3), respectively. *(iv)* **Our method effectively balances all three target properties, outperforming the previous SOTA method, MEND**. Compared to MEND which transforms the gradients of knowledge editing to a generalizable direction for keeping both locality and generality, UniKE significantly achieves superior locality (93.8 vs. 89.4 on average) and generality (95.1 vs. 88.6 on average).

Table 2: Main results on sequential editing on the MMEdit.

(a) Results on 10-step editing on VQA.

| Method | Rel. | T-Gen. | M-Gen. | T-Loc. | M-Loc. |
|---|---|---|---|---|---|
| FT | 67.8 | 62.4 | 58.3 | 66.9 | 62.3 |
| KE | 83.2 | 82.1 | 57.9 | 84.6 | 64.3 |
| T-Patcher | 79.2 | 62.5 | 61.4 | 83.4 | 79.5 |
| MEND | 90.6 | 86.3 | 79.5 | 87.4 | 76.1 |
| SERAC | 87.4 | 84.4 | 82.7 | 87.9 | 12.5 |
| **UniKE (Ours)** | **91.5** | **87.1** | **85.4** | **88.9** | **83.1** |

(b) Results on 20-step editing on VQA.

| Method | Rel. | T-Gen. | M-Gen. | T-Loc. | M-Loc. |
|---|---|---|---|---|---|
| FT | 64.9 | 59.6 | 57.9 | 62.6 | 61.7 |
| KE | 79.8 | 74.3 | 55.7 | 80.0 | 60.1 |
| T-Patcher | 76.6 | 56.6 | 54.8 | 81.5 | 78.2 |
| MEND | 85.1 | 80.4 | 75.7 | 82.2 | 73.9 |
| SERAC | 87.1 | 83.0 | 81.0 | 85.5 | 10.7 |
| **UniKE (Ours)** | **88.9** | **83.4** | **81.2** | **85.7** | **79.6** |

(c) Results on 10-step editing on image caption.

| Method | Rel. | T-Gen. | M-Gen. | T-Loc. | M-Loc. |
|---|---|---|---|---|---|
| FT | 65.3 | 63.8 | 61.9 | 58.9 | 49.8 |
| KE | 84.3 | 64.2 | 54.3 | 90.0 | 60.1 |
| T-Patcher | 80.7 | 67.5 | 63.6 | 89.5 | 80.6 |
| MEND | 90.2 | 89.6 | 73.5 | 90.9 | 73.7 |
| SERAC | 88.0 | 87.6 | 86.7 | 92.1 | 6.8 |
| **UniKE (Ours)** | **91.8** | **90.4** | **89.1** | **93.5** | **85.0** |

(d) Results on 20-step editing on image caption.

| Method | Rel. | T-Gen. | M-Gen. | T-Loc. | M-Loc. |
|---|---|---|---|---|---|
| FT | 64.0 | 63.1 | 60.6 | 56.4 | 48.2 |
| KE | 80.0 | 58.8 | 51.2 | 84.3 | 57.9 |
| T-Patcher | 76.0 | 64.1 | 60.0 | 88.8 | 79.8 |
| MEND | 85.0 | 84.0 | 71.4 | 87.3 | 71.1 |
| SERAC | 87.4 | 87.0 | **85.6** | **90.3** | 4.8 |
| **UniKE (Ours)** | **88.4** | **87.2** | **85.6** | 90.1 | **81.1** |

Table 4: Results of ablation study to illustrate the effect of individual components.

| | EDITING VQA (E-VQA) | | | | | EDITING IMAGE CAPTION (E-IC) | | | | |
|---|---|---|---|---|---|---|---|---|---|---|
| Model | Rel. ↑ | T-Gen. ↑ | M-Gen. ↑ | T-Loc. ↑ | M-Loc. ↑ | Rel. ↑ | T-Gen. ↑ | M-Gen. ↑ | T-Loc. ↑ | M-Loc. ↑ |
| 1 *only* Intrin | 83.5 | 69.2 | 67.4 | 85.3 | 83.1 | 85.7 | 73.3 | 68.5 | 94.1 | 84.4 |
| 2 *only* Latent-IKE | 94.6 | 92.2 | 93.3 | 54.1 | 30.5 | 89.6 | 85.9 | 84.4 | 59.0 | 36.8 |
| 3 Intrin+IKE | 95.5 | 79.0 | 71.8 | 63.8 | 50.1 | 89.9 | 77.2 | 74.4 | 61.0 | 59.4 |
| 4 Intrin+Latent-IKE | 95.9 | 93.2 | 89.6 | 95.2 | 85.3 | 96.5 | 92.4 | 89.7 | 95.2 | 85.5 |
| **UniKE (Ours)** | **98.0** | **97.4** | **92.8** | **98.7** | **88.8** | **96.8** | **95.7** | **92.4** | **98.9** | **87.3** |

## 4.3 Main Results on Sequential Editing

In $K$-step sequential editing, the model is sequentially edited while encountering mistakes in $\mathcal{D}_{edit}(|\mathcal{D}_{edit}| = K)$. After the $K$th edit, the post-edit MLLM is utilized to evaluate the target properties. Table 2 shows the results of sequential editing ($K = 10, 20$; we exclude IKE as its setup in sequential editing is meaningless). It can be observed that *(i)* whether in editing VQA or image captioning tasks, there is a significant decline in the performance of most methods as the number of editing steps ($K$) increases. Particularly for MEND, while it remains competitive in one-step editing, the results of sequential editing are suboptimal. *(ii)* The performance of external knowledge resorting (SERAC) is minimally affected by the increase in $K$. However, it inherently suffers from a lack of locality. *(iii)* In contrast, our method **consistently maintains superior performance compared to the baselines**. It consistently outperforms MEND across all metrics of sequential editing, **with its advantages over the baseline becoming increasingly significant as $K$ increases.**

## 4.4 Main Results on Cross-task Editing

Cross-task editing builds on the foundation of sequential editing (we select $K = 10$) and requires the MLLM to simultaneously edit VQA and image-caption samples within the same sequence. Table 3 presents the results of cross-task editing (averaging the results over all E-VQA and E-IC samples). It is evident that most baselines struggle to effectively edit both tasks within a single editing sequence. In contrast, UniKE excels at integrating the knowledge from these two distinct tasks, significantly outperforming baseline methods in terms of reliability, generality, and locality.

Table 3: Main results on cross-task editing.

| Method | Rel. | T-Gen. | M-Gen. | T-Loc. | M-Loc. |
|---|---|---|---|---|---|
| FT | 65.0 | 63.2 | 59.4 | 57.3 | 52.2 |
| KE | 83.0 | 71.3 | 56.0 | 85.5 | 60.2 |
| T-Patcher | 79.0 | 63.2 | 61.2 | 84.0 | 79.8 |
| MEND | 88.8 | 87.4 | 75.3 | 88.1 | 73.6 |
| SERAC | 87.5 | 85.0 | 83.1 | 90.0 | 6.6 |
| **UniKE** | **90.7** | **88.2** | **86.8** | **90.4** | **83.8** |

## 4.5 In-Depth Analysis

**Effect of Individual Components.** We investigate the effectiveness of each component and conduct the following experiments on one-step editing: **(1)** *only* **Intrin &** *only* **Latent-IKE:** We utilize either intrinsic knowledge editing or external knowledge resorting (Latent IKE), conducting multimodal editing in the latent space. In Rows 1 and 2 of Table 4, it is evident that single-type knowledge editing approaches cannot simultaneously possess all three properties well, resulting in either generality or locality being unsatisfactory. **(2) Intrin + IKE:** We simply combine intrinsic knowledge editing and vanilla in-context editing without paradigm unification. The results in Row 3 demonstrate that if

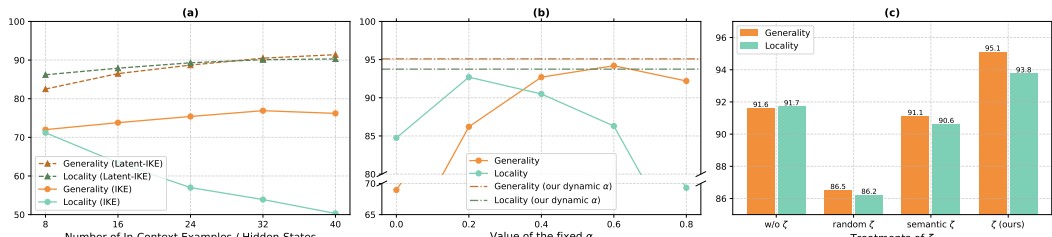

Figure 4: **(a)** Performance for IKE & Latent-IKE (both combined with intrinsic knowledge editing) with different number of in-context examples or hidden states on E-VQA. **(b)** Performance with different fixed value of $\alpha$ and our dynamic $\alpha$. **(c)** Performance with different $\zeta$ treatments.

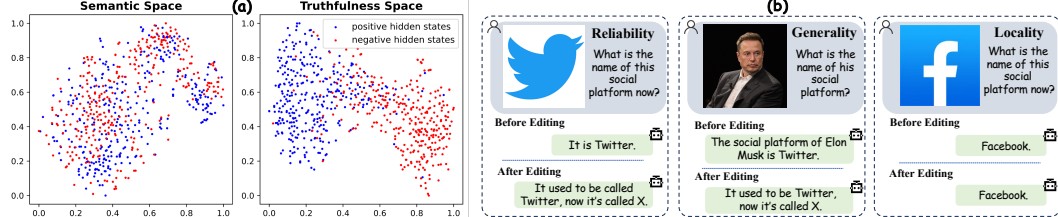

Figure 5: **(a)** Visualization of different knowledge spaces. **(b)** A qualitative example.

integrating both types of knowledge editing methods without a unified paradigm, it is difficult to fully leverage the individual advantages of each method, still leading to suboptimal generality and locality. **(3) Intrin + Latent IKE:** We remove the enhanced knowledge collaboration proposed in §3.3. The results in Row 5 validate that the enhanced collaboration based on knowledge representation disentangling further enables the post-edit MLLM to achieve superior generality and locality.

**Effect of In-context Editing in Latent Space.**   To validate the superiority of in-context editing in latent space, in Figure 4.a we compare Latent IKE with vanilla IKE across different numbers of in-context samples/hidden-states (both combined with intrinsic knowledge editing without enhanced knowledge collaboration). IKE is criticized that in-context samples are difficult to quantitatively control and take up context window space [27, 23]. It can be observed that in IKE, as the number of in-context samples increases, though generality generally trends upward, there is a notable decline in locality. While in our method, via quantitatively controlling the inclusion of in-context hidden states while also reducing the prompt length, **both generality and locality of the post-edit MLLM show an overall upward trend as the number of in-context samples increases**.

**Effect of Knowledge Collaboration in Semantic Space.**   During knowledge collaboration, in semantic space, we adaptively adjust the inclusion magnitude of in-context knowledge (assigning the cosine similarity between $h_{know}^{sem}$ and $\mathtt{h}_{input}[-1]$ to $\alpha$). To demonstrate the superiority of this strategy, we further experiment with several fixed values for $\alpha$. As shown in Figure 4.b, the increase in the fixed $\alpha$ enhances the impact of in-context knowledge, which tends to improve generality. However, it also leads to a reduction in locality. In contrast, our method, which adaptively adjusts $\alpha$ based on semantic relevance, customizes an appropriate injection weight for each in-context knowledge. **Thereby, we ensure an enhancement in generality while also preventing the disruption to locality.**

**Effect of Knowledge Collaboration in Truthfulness Space.**   During knowledge collaboration, in the truthfulness space, we identify a truthful editing direction, $\zeta$ to guide intrinsic knowledge editing. To assess the effects of $\zeta$, we conduct the following experiments: **(1) w/o $\zeta$:** removing the regulation of $\zeta$ as per Eq.(3); **(2) random $\zeta$:** replacing $\zeta$ with a random tensor; **(3) semantic $\zeta$:** generating $\zeta$ in the same manner within the semantic space. As depicted in Figure 4.c, **$\zeta$ could further enhance the generalizability of intrinsic knowledge, thus achieving superior editing performance.** However, when $\zeta$ does not point towards the correct editing direction (random or semantic $\zeta$), it acts as a disruptor, thereby impairing the editing performance compared to w/o $\zeta$.

**Visualization of Different Knowledge Spaces.**   To give an intuitive perspective on the disentangled knowledge representations, we employ t-SNE [37] for dimensionality reduction, visualizing the embedding distributions for semantic and truthfulness representations across both positive and

negative hidden states. As shown in Figure 5.a, the positive and negative hidden states display similar distributions in the semantic space, yet are distinctly separated in the truthfulness space. This visualization effectively confirms the efficacy of our approach to knowledge disentangling.

**Qualitative Examples.** As shown in Figure 5.b and Appendix E, UniKE achieves reliable multi-modal editing while generalizing to similar scenarios and ensuring accuracy for irrelevant examples.

## 5 Conclusion

In this paper, we introduce UniKE, a multimodal editing framework that establishes a unified paradigm for both intrinsic knowledge editing and external knowledge resorting. We conceptualize both types of knowledge as vectorized key-value memories and effectively enhance their collaborations with knowledge disentangling. Extensive experimental results demonstrate that our method enhances the post-edit MLLMs across various settings (one-step editing, sequential editing, and cross-task editing), ensuring that they maintain excellent reliability, generality, and locality simultaneously.

## Acknowledgements

This work has been supported in part by the Key Research and Development Projects in Zhejiang Province (No. 2024C01106, 2024C01102), the NSFC (No. 62272411), the National Key Research and Development Project of China (2018AAA0101900).

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

# Appendix

## A In-context Knowledge as External Key-Value Memory

In §3.1, we provide a brief analysis of how in-context knowledge can be considered as an external vectorized key-value memory, and how to conduct in-context editing as feature shifting in the latent space. Here, we present a more detailed derivation of Eq.(2). Specifically, in-context learning typically concatenates the external multimodal knowledge $X_{know}$ with the original input sequence $X_{input}$ to form the combined sequence $X = [X_{know}, X_{input}]$. Considering the self-attention mechanism $\texttt{Attn}(Q, K, V) = \text{softmax}(QW_qW_k^\top K^\top)VW_v$ in the transformer ($W_k, W_q, W_v$ denote the learnable key, query, and value matrices, respectively), the in-context knowledge simply changes the attention module through prepending a context matrix before the original input examples. During in-context learning, the attention $\texttt{Attn}(Q = X_{input}, K = X, V = X)$ for tokens in the original input sequence can actually be formulated as follows:

$$
\begin{aligned}
&\text{Attn}(X_{input}, X, X) \\
&= \text{Attn}(X_{input}, [X_{know}, X_{input}], [X_{know}, X_{input}]) \\
&= \text{Softmax}\left(X_{input}W_q\left(X_{know}W_k, X_{input}W_k\right)^\top\right)\begin{pmatrix}X_{know}\\X_{input}\end{pmatrix}W_v \\
&= \alpha \cdot \text{Softmax}\left(X_{input}W_qW_k^\top X_{input}^\top\right)X_{input}W_v + (1-\alpha)\cdot\text{Softmax}\left(X_{input}W_qW_k^\top X_{know}^\top\right)X_{know}W_v \\
&= \alpha \cdot \text{Attn}(X_{input}, X_{input}, X_{input}) + (1-\alpha)\cdot\text{Attn}(X_{input}, X_{know}, X_{know}) \\
&= \alpha \cdot \mathtt{h}_{input} + (1-\alpha)\cdot\mathtt{h}_{know},
\end{aligned}
\tag{8}
$$

Here $\alpha$ is a scalar that represents the sum of normalized attention weights between in-context knowledge and the original input:

$$
\alpha = \frac{\sum_i \exp(x_{input}W_qW_k^\top X_{input}^\top)_i}{\sum_i \exp(x_{input}W_qW_k^\top X_{know}^\top)_i + \sum_j \exp(x_{input}W_qW_k^\top X_{input}^\top)_j}
\tag{9}
$$

where $x_{input}$ is the token within the original input sequence. We can find that in-context learning actually applies a position-wise modification to the original attention output by shifting the original output features, with the self-attention controlling the shift direction and $\alpha$ controlling the shift distance. Therefore, in this paper, we dynamically adjust the value of $\alpha$ to control the inclusion magnitude of in-context knowledge representations.

## B Task Definition

### B.1 Multimodal Knowledge Editing

The goal of multimodal knowledge editing is to efficiently modify an initial MLLM's behavior based on a specific edit descriptor, without incurring significant retraining costs or affecting its behavior on other unrelated samples. Formally, $(v_e, x_e, y_e) \in \mathcal{D}_{edit}$ is the edit descriptor, where $v_e$ refers to the visual input, $x_e$ refers to the textual input, $y_e$ denotes the desired output. And we represent the MLLM (with its parameters denoted as $\theta$) as a function $f : (\mathbb{V}, \mathbb{X}) \longrightarrow \mathbb{Y}$ that maps the multimodal input $(v, x)$ to its corresponding prediction $y_o = f_\theta(v, x)$. For intrinsic knowledge editing, the parameters of the post-edit MLLM $f^{post}$ are updated to $\theta_e((v_e, x_e, y_e))$; while for external knowledge resorting methods, the MLLM's parameters remain $\theta$, and additional relevant external knowledge $\mathcal{K}$ is incorporated as the extra input. On this basis, a successful edit should first adjust the MLLM's output on the input $(v_e, x_e)$ from $y_o$ to $y_e$. Additionally, there is a broad set of inputs closely associated with the edit descriptor, referred to as the editing neighbor $\mathcal{N}(v_e, x_e)$. The MLLM's behavior should also be corrected for examples within this neighbor while maintaining its performance for out-of-neighbor examples unaltered:

$$
f^{post}(v, x) = f_{\theta_e}(v, x) \texttt{ or } f_\theta(v, x, \mathcal{K}) = \begin{cases} y_e & \text{if } (v, x) \in \mathcal{N}(v_e, x_e) \\ y_o & \text{if } (v, x) \notin \mathcal{N}(v_e, x_e) \end{cases}
\tag{10}
$$

Based on the above analysis, there are often three metrics used to measure the performance of the post-edit MLLM: reliability, locality, and generality.

**Reliability.** Knowledge editing is reliable when the post-edit MLLM successfully changes prediction from $y_o$ to $y_e$ [11]. We access the reliability based on the average accuracy of the edit case:

$$\mathcal{M}_{rel} = \mathbb{E}_{(v_e, x_e, y_e) \sim \mathcal{D}_{\text{edit}}} \left[ \mathbb{1}_{f(v_e, x_e; \theta_e(v_e, x_e, y_e)) = y_e} \right] \tag{11}$$

**Locality.** Knowledge editing should be implemented locally, ensuring that the post-edit MLLM should not change the output of irrelevant out-of-neighbor examples. And we leverage two metrics to evaluate locality: $\mathcal{M}_{loc}^{Text}$ (**T-Locality**) and $\mathcal{M}_{loc}^{Img}$ (**M-Locality**). For T-Locality, we remove the visual modules of MLLM, and leverage rudimentary question-and-answer datasets $\mathcal{D}_{\text{Loc-T}} = \{(x_t, y_t)\}$ to examine whether the MLLM's understanding of pure textual input remains unaffected.

$$\mathcal{M}_{loc}^{Text} = \mathbb{E}_{\substack{(v_e, x_e, y_e) \sim \mathcal{D}_{\text{edit}} \\ (x_t, y_t) \sim \mathcal{D}_{\text{Loc-T}}}} \left[ \mathbb{1}_{f(x_t; \theta_e(v_e, x_e, y_e)) = f(x_t, \theta)} \right] \tag{12}$$

Of course, we also need to consider the potential ramifications of knowledge editing on visual comprehension. Given $\mathcal{D}_{Loc-M} = \{(v_m, x_m, y_m)\}$, M-Locality is measured by the rate at which the post-edit MLLM maintains the same predictions as the pre-edit MLLM on multimodal input.

$$\mathcal{M}_{loc}^{Img} = \mathbb{E}_{\substack{(v_e, x_e, y_e) \sim \mathcal{D}_{\text{edit}} \\ (v_m, x_m, y_m) \sim \mathcal{D}_{\text{Loc-M}}}} \left[ \mathbb{1}_{f(v_m, x_m; \theta_e(v_e, x_e, y_e)) = f(v_m, x_m; \theta)} \right] \tag{13}$$

**Generality.** It is not sufficient for knowledge editing to merely correct individual erroneous inputs. The post-edit MLLM should also generalize to equivalent neighbors with strong generalization [14, 32]. In multimodal scenarios, equivalent neighbors can be rephrased textual sentences or rephrased images, corresponding to the metrics of T-Generality and M-Generality, respectively. And then generality is assessed by the average accuracy on examples uniformly sampled from these equivalent neighbors ( $\mathcal{N}(x_e)$ or $\mathcal{N}(v_e)$ ).

$$\mathcal{M}_{gen}^{Text} = \mathbb{E}_{\substack{(v_e, x_e, y_e) \sim \mathcal{D}_{\text{edit}} \\ (x_r) \sim \mathcal{N}(x_e)}} \left[ \mathbb{1}_{f(v_e, x_r; \theta_e) = y_e} \right] \tag{14}$$

$$\mathcal{M}_{gen}^{Img} = \mathbb{E}_{\substack{(v_e, x_e, y_e) \sim \mathcal{D}_{\text{edit}} \\ (v_r) \sim \mathcal{N}(v_e)}} \left[ \mathbb{1}_{f(v_r, x_e; \theta_e) = y_e} \right] \tag{15}$$

## B.2 Sequential Editing

Previous multi-modal editing focuses on one-step editing, addressing a single error from one target sample at a time, and subsequently evaluating the three metrics based on the sample itself, its equivalent neighbors, and its out-of-neighbor examples. However, one-step editing is not applicable to practical situations. Following [11], we extend multimodal editing into the setup of sequential editing. In $K$-step sequential editing, we have a set of target samples $\mathcal{D}_{\text{seq}}$ to be edited in a sequence ($|\mathcal{D}_{\text{seq}}| = K$; each sample in $\mathcal{D}_{\text{seq}}$ also contains its own equivalent neighbors and out-of-neighbor examples). After continuously editing all $K$ target samples, we obtain a post-edit MLLM $f_{\theta_e\left(\sum_{(v,x,y) \in \mathcal{D}_{\text{seq}}}(v,x,y)\right)}$.

We first evaluate the reliability, locality, and generality of the post-edit MLLM within the sequence in a manner similar. And the final performance of the above metrics will be averaged over all $\mathcal{D}_{\text{seq}} \sim \mathcal{D}_{\text{edit}}$.

$$\mathcal{M}'_{rel} = \mathbb{E}_{\mathcal{D}_{\text{seq}} \sim \mathcal{D}_{\text{edit}}} \left[ \frac{1}{K} \sum_{(v_e, x_e, y_e) \in \mathcal{D}_{\text{seq}}} \mathbb{1}_{f\left(v_e, x_e; \theta_e\left(\sum_{(v,x,y) \in \mathcal{D}_{\text{seq}}}(v,x,y)\right)\right) = y_e} \right] \tag{16}$$

$$\mathcal{M}'^{Text}_{loc} = \mathbb{E}_{\mathcal{D}_{\text{seq}} \sim \mathcal{D}_{\text{edit}}} \left[ \frac{1}{K} \sum_{(x_t, y_t) \in \mathcal{D}_{\text{Loc-T}}} \mathbb{1}_{f\left(x_t; \theta_e\left(\sum_{(v,x,y) \in \mathcal{D}_{\text{seq}}}(v,x,y)\right)\right) = f(x_t, \theta)} \right] \tag{17}$$

$$\mathcal{M}'^{Img}_{loc} = \mathbb{E}_{\mathcal{D}_{\text{seq}} \sim \mathcal{D}_{\text{edit}}} \left[ \frac{1}{K} \sum_{(v_m, x_m, y_m) \in \mathcal{D}_{\text{Loc-M}}} \mathbb{1}_{f\left(v_m, x_m; \theta_e\left(\sum_{(v,x,y) \in \mathcal{D}_{\text{seq}}}(v,x,y)\right)\right) = f(v_m, x_m, \theta)} \right] \tag{18}$$

$$\mathcal{M}'^{Text}_{gen} = \mathbb{E}_{\mathcal{D}_{\text{seq}} \sim \mathcal{D}_{\text{edit}}} \left[ \frac{1}{K} \sum_{(v_e, x_e, y_e) \in \mathcal{D}_{\text{seq}}, (x_r) \sim \mathcal{N}(x_e)} \mathbb{1}_{f\left(v_e, x_r; \theta_e\left(\sum_{(v,x,y) \in \mathcal{D}_{\text{seq}}}(v,x,y)\right)\right) = y_e} \right] \tag{19}$$

$$\mathcal{M}'^{Img}_{gen} = \mathbb{E}_{\mathcal{D}_{\text{seq}} \sim \mathcal{D}_{\text{edit}}} \left[ \frac{1}{K} \sum_{(v_e, x_e, y_e) \in \mathcal{D}_{\text{seq}}, (v_r) \sim \mathcal{N}(v_e)} \mathbb{1}_{f\left(v_r, x_e; \theta_e\left(\sum_{(v,x,y) \in \mathcal{D}_{\text{seq}}}(v,x,y)\right)\right) = y_e} \right] \tag{20}$$

### B.3 Cross-Task Editing

On the basis of sequential editing, we further allow the $K$ target samples within a sequence to come from different tasks. For example, a single sequence might include samples from both the VQA task and the image caption task. This requires the MLLM to integrate knowledge from distinct tasks to achieve cross-task editing. The evaluation for the above metrics is similar to that of sequential editing. However, instead of separately reporting metrics for each task, we mix the data from different tasks and calculate a single set of (Reliability, T-Locality, M-Locality, T-Generality, M-Generality) for the mixed data. This provides a measure of the post-edit MLLM's average performance across different tasks.

## C Preparing In-context Knowledge for Representation Extraction

To extract representations of in-context knowledge, we should first prepare a set of in-context knowledge for the MLLM. In multimodal scenarios [18, 17, 44, 2, 9, 51, 49, 39, 21, 20, 22, 12, 38, 7], We aim to provide in-context knowledge that the MLLM has not previously mastered, which means the MLLM cannot provide correct answers to the corresponding questions.

And we attribute MLLMs' hallucinated responses to deficiencies at three levels, as shown in Figure 6: **(1) Insufficient vision extraction.** MLLMs first utilizes a Visual Prompt Generator (VPG, *e.g.*, Qformer [19]) to abstract image features [48]. However, some necessary reasoning-aware visual details that complement the primary content and semantically connect the text instructions, may be ignored and not extracted by the VPG [16, 15]. **(2) Inaccurate vision recognition.** Even if the VPG extracts sufficient visual features for reasoning, the MLLM might fail to understand the corresponding visual objects if it has not been sufficiently exposed to these feature patterns during training, resulting in inaccurate recognition of visual objects [24, 43]. **(3) Incorrect text-vision collaborative reasoning.** Even with sufficient vision extraction and accurate vision recognition, MLLMs may struggle with understanding the

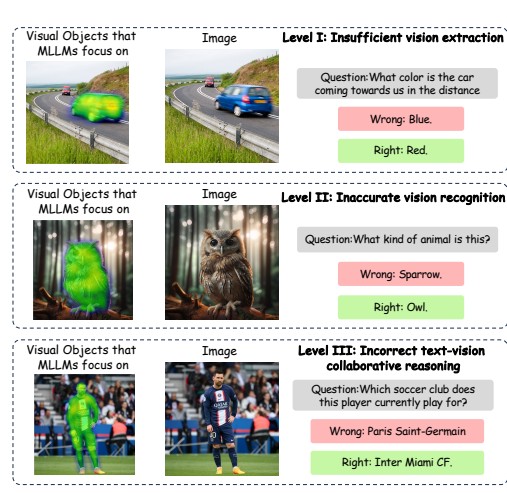

Figure 6: We attribute MLLMs' hallucinated responses to deficiencies at three levels.

spatial relationships between different visual regions. Moreover, they may exhibit errors in common-sense knowledge when combining vision with text instructions, ultimately resulting in incorrect text-vision collaborative reasoning [42].

On this basis, we collect a substantial set of hallucinated predictions from MLLMs across these three levels to develop in-context knowledge. For each multimodal question $Q_I$ with MLLMs' hallucinated predictions $A_{neg}$, we additionally provide a truthful answer $A_{pos}$. For each piece of knowledge, we pair $Q_I + A_{neg}$ as the negative knowledge and $Q_I + A_{pos}$ as the positive knowledge. For example, assuming [IMG] is an image of Lionel Messi, "[IMG] *Question: Which soccer club is this player currently playing for? Answer: Paris Saint-Germain.*" represents the negative knowledge; while "[IMG] *Question: Which soccer club is this player currently playing for? Answer: Inter Miami.*" represents the positive knowledge. Subsequently, as stated in §3.3, we can extract several critical hidden states for the collected in-context knowledge.

## D Experimental Details

### D.1 Dataset

We conduct experiments on the MMEdit benchmark [4], which consists of two sub-tasks: Editing VQA (E-VQA) and Editing Image Caption (E-IC). We leverage Reliability, locality (T-Locality and M-Locality), and Generality (T-Generality and M-Generality) as the evaluation metrics. The definitions of each metric are given in the previous section (Appendix B). We leverage BLIP2-

Table 5: Average results of one-step / sequential editing across various sub-tasks, backbones, and sub-metrics (locality and generality, with T-Locality&M-Locality and T-Locality&T-Generality as the specific sub-metrics). The best result is marked **bold**. The second best result is underlined.

| Methods | ONE-STEP EDITING | | | SEQUENTIAL EDITING | | |
|---|---|---|---|---|---|---|
| | Reliability | Generality | Locality | Reliability | Generality | Locality |
| FT | 64.3 | 58.1 | 61.2 | 65.5 | 61.0 | 58.4 |
| KE | 81.1 | 65.4 | 72.4 | 81.8 | 62.3 | 72.7 |
| T-Patcher | 84.5 | 71.6 | 87.2 | 78.1 | 61.3 | 82.7 |
| MEND | **98.6** | 88.6 | 89.4 | 87.7 | 80.0 | 80.3 |
| IKE | 96.8 | 88.6 | 26.8 | – | – | – |
| SERAC | 94.7 | 91.3 | 52.5 | 87.5 | 84.8 | 48.8 |
| **UniKE (Ours)** | 98.0 | **95.1** | **93.8** | **90.2** | **86.2** | **85.9** |

OPT [19] and MiniGPT-4 [48] as the backbone models, which are under BSD 3-Clause License. And the MMEdit benchmark is under MIT license.

Moreover, the original setup of MMEdit only involves one-step editing, where each edit aims to correct a single mistake from a single target sample, and the above metrics are assessed after each edit. We further extend the setup to sequential editing and cross-task editing, both of which are defined in the previous section. For sequential editing in E-VQA and E-IC, we select $K = 10$ and $K = 20$ as the editing steps within a sequence. For cross-task editing, we choose $K = 10$, with each editing sequence containing 5 E-VQA samples and 5 E-IC samples. Additionally, we no longer report the results for E-IC and E-VQA separately in cross-task editing; instead, we present the average results of all E-IC and E-VQA samples.

## D.2   Baselines

**Fine-tune.**   Fine-tuning is the most widely employed strategy for adapting pre-trained language models to specific tasks. So we leverage vanilla fine-tuning as the baseline for multimodal editing. We only tune the last layer of MLLM, which has been verified as the most effective fine-tuning strategy in [4].

**Knowledge Editor (KE).**   KE [6] is a intrinsic knowledge editing method that corrects erroneous knowledge in language models without re-training the whole model. It leverages a hypernetwork (a bidirectional-LSTM) to predict the weight update for constrained optimization.

**MEND.**   Model Editor Networks with Gradient Decomposition (MEND [28]) is also a method of intrinsic knowledge editing. It learns to transform the editing gradients into a generalizable direction via employing a low-rank decomposition of gradients, **aiming to keep both generality and locality during knowledge editing**.

**T-Patcher.**   T-Patcher [11] is also a typical intrinsic knowledge editing method that integrates addition neurons for addressing mistakes in the last several layers of the Feed-Forward Network (FFN) within language models.

**In-context Knowledge Editing (IKE).**   IKE [46] edits the language model by prompting the model with several retrieved edit demonstrations from the external database. As a result, the language model can generate outputs that align with the provided knowledge when given a refined knowledge context as a prompt.

**SERAC.**   SERAC [29] is also a method of external knowledge resorting. It leverages an explicit memory system to cache edits, which is later utilized to adjust the output of the language model during inference. Moreover, the memory system employs a small auxiliary scope classifier to determine whether the input falls within the scope of the memory cache.

## D.3   Implementation Details

We conduct knowledge editing in the latent space with a unified paradigm. In intrinsic knowledge editing, we add extra 10 key-value pairs in the FFN of the last four transformer layers; for external

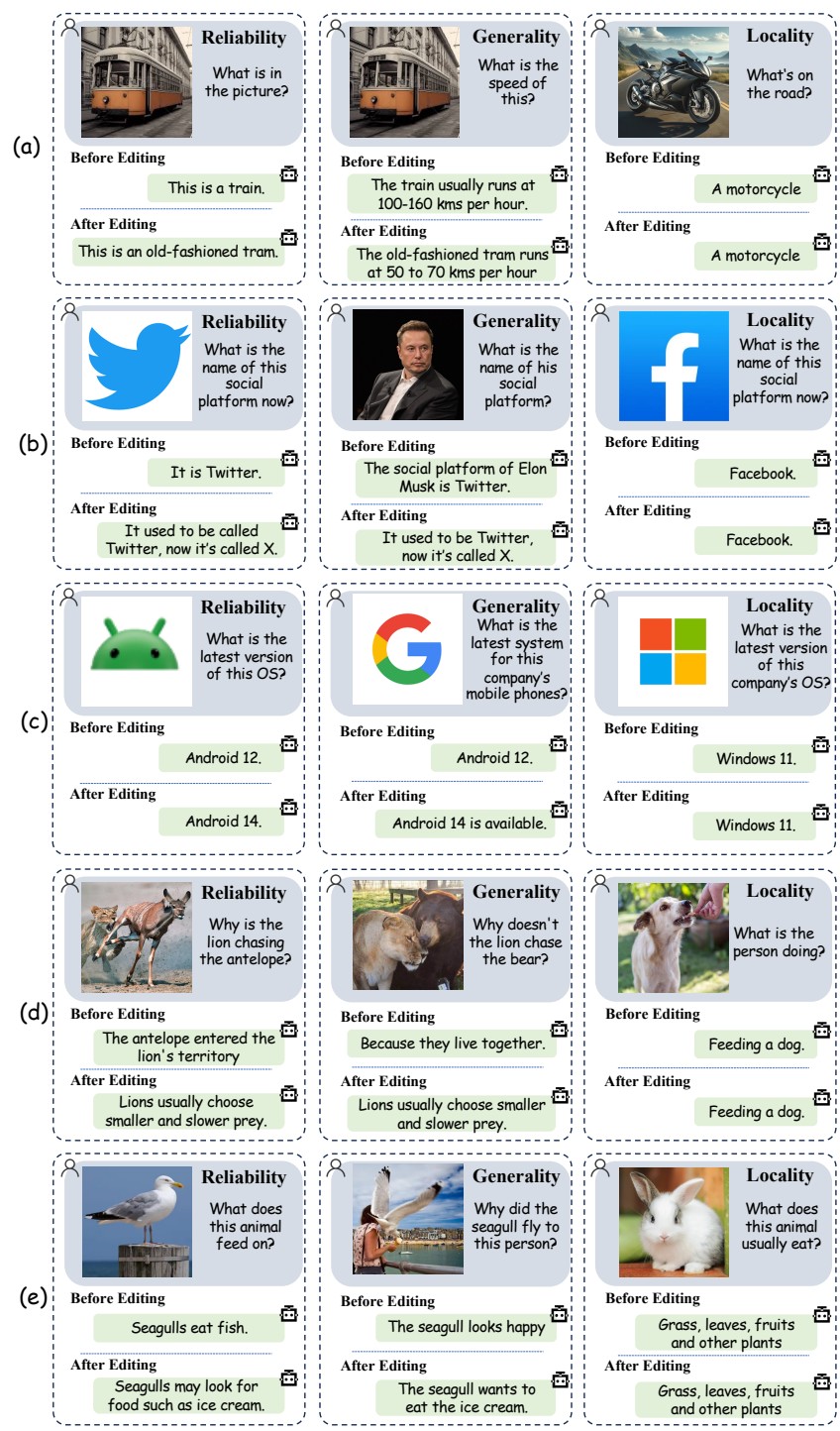

Figure 7: Qualitative examples of multimodal editing.

knowledge resorting, we retrieve top-40 hidden states of in-context knowledge with the highest similarity for each case and conduct feature shifting for in-context editing in the last four transformer layers. To extract the hidden states of in-context knowledge, we collect more than 15K triplets. Furthermore, during knowledge disentangling, both the truthfulness encoder and the semantic encoder simply consist of several MLP layers. We leverage the collected in-context knowledge representations to pre-train the encoders via contrastive learning [50]. During contrastive learning, both encoders are optimized using the Adam optimizer with a learning rate of 1e-4. After that, completing a single

one-step edit takes only a matter of seconds and we run all experiments with 6 NVIDIA RTX A6000 GPUs.

### D.4 Statistical Summary of Results

In Table 5, we present a statistical summary of the average reliability, locality, and generality of the post-edit MLLM for both one-step and sequential editing. Although some methods (*e.g.*, MEND) achieve reliable editing in multimodal scenarios, UniKE effectively balances all three target properties, significantly outperforming baseline methods in terms of generality and locality.

## E  Qualitative Examples

In Figure 7, we show more qualitative examples. In case (a), after altering the MLLM's cognitive bias of a particular concept through knowledge editing, we enable the MLLM to leverage its existing world knowledge to **automatically correct other related details** associated with that concept. In cases (b) and (c), after updating outdated knowledge through UniKE, the MLLM can **activate this updated knowledge** to correctly answer a series of related questions. In cases (d) and (e), model editing enables the MLLM to **look beyond the superficial aspects and answer questions from a deeper perspective**. Furthermore, for all five cases, we ensure the locality of the post-edit MLLM without altering its original responses to irrelevant inputs.

## F  Limitations

There still exist some limitations in our work: (1) At present, we have only considered editing MLLMs with visual comprehension capabilities, and have not addressed editing in visual generation scenarios. (2) Due to the resource limitation, we do not afford to edit MLLMs with a larger number of parameters such as the 65B LLaMA Adapter V2 [10].

## G  Broader Impacts

**Ethical Impacts.** This study does not raise any ethical concerns. The research does not involve subjective assessments or the use of private data. Only publicly available datasets and models are utilized for experimentation.

**Expected Societal Implications**. This study proposes a data- and time-efficient way to edit MLLMs. A major societal concern with this technology lies in its potential for misuse. For example, some malicious individuals may exploit our technology to fabricate false information for knowledge editing and spread rumors. To counter these threats, it is crucial to develop strong ethical standards and implement ongoing surveillance.

