# OpenReview forum: "Towards Unified Multimodal Editing with Enhanced Knowledge Collaboration"
_NeurIPS.cc/2024/Conference — NeurIPS 2024 spotlight_

### Official Review · Reviewer_4G2a · 2024-07-06

**Soundness:** 3
**Presentation:** 2
**Contribution:** 3
**Rating:** 6
**Confidence:** 4

**Summary:**

The paper introduces UniKE, a novel multimodal editing method that addresses challenges in knowledge editing for Multimodal Large Language Models. UniKE unifies intrinsic knowledge editing and external knowledge resorting by vectorized key-value memories. By disentangling knowledge representations into semantic and truthfulness spaces, UniKE promotes collaboration between intrinsic and external knowledge editing, enhancing the post-edit MLLM's reliability, generality, and locality. Extensive experiments validate UniKE's effectiveness.

**Strengths:**

1.	UniKE establishes a unified framework for intrinsic knowledge editing and external knowledge resorting.
2.	Extensive experiments demonstrate that UniKE consistently maintains excellent reliability, generality, and locality

**Weaknesses:**

1.	The presentation is confusing. For example, in intrinsic knowledge editing, how do you actual edit the intrinsic knowledge in FFN. Do you follow the same pipeline as T-Patcher or adding an additional neural network? Although intrinsic knowledge is considered as key-value pairs but they are different in the end.
2.	The unified framework and disentanglement of knowledge representations into semantic and truthfulness spaces introduces additional complexity to the editing process. If I understand correctly from the implementation details, it requires more than 15k additional triplets to train the encoders.
3.	In the experiments, I think /beta is another hyperparameter, but I did not find an experiment discussing it.

**Questions:**

Why would you add extra 10 key-value pairs in the FFN of the last four transformer layers in intrinsic knowledge editing?
It seems that \zeta is quite sensitive. Could you explain that?

---

> ### Author Rebuttal · Authors · 2024-08-07
>
> We sincerely appreciate your constructive comments. We will explain your concerns point by point.
>
> **Q1:** The presentation is confusing. For example, in intrinsic knowledge editing, how do you actual edit the intrinsic knowledge in FFN?
>
> **A1:** We apologize if our manuscript causes any confusion. **The intrinsic knowledge editing is implemented by adding 10 knowledge neurons in the FFN of the last four transformer layers** with a similar pipeline as T-patcher.
> The addition of neurons can be understood as an expansion of the dimension of the weight matrix in the FFN, as shown in Eq.3.
> This process retains the original $d\times d'$
> weights in the FFN while appending a learnable parameter of dimensions $d\times ne$ as the new neurons.
> And Eq.2 demonstrates why intrinsic knowledge can be viewed as key-value pairs in this context.
>
> Although this pipeline may seem similar to TPatcher, **the key innovation of our paper lies in consistently transforming in-context editing to the latent level and integrating it into the self-attention**, which allows both external and intrinsic knowledge editing to **follow a unified paradigm within a cohesive framework**. Furthermore, **our intrinsic knowledge editing is actually an improvement of T-Patcher**. By synergistically unifying the paradigms of both editing methods and incorporating knowledge disentangling, we identify a truthful editing direction, $\zeta$, which **further guides intrinsic knowledge into a more generalizable direction**. To understand how $\zeta$ enhances intrinsic knowledge editing, please refer to Eq.6, which is inspired by the success of [Chen et al 2022].
>
> [Chen et al 2022]AdaptFormer: Adapting Vision Transformers for Scalable Visual Recognition.Neurips2022.
>
> &nbsp;
>
> **Q2:** The unified framework and knowledge disentanglement introduce additional complexity to the editing.
>
> **A2:** Thank you for valuable feedback. We would like to address your concerns from two points.
>
> **(2.1)** The unified framework and knowledge disentanglement almost **incur no additional time or GPU memory overhead** during the editing process. Both the truthfulness and semantic encoders consist of several MLP layers **with only 4M parameters** (MLLM contains 7B parameters), which **remain frozen** during editing. As shown in **Table 10 in the Rebuttal PDF, these encoders enhance editing performance with NEGLIGIBLE additional time and memory overhead**.
> You can also refer to **Table 7 in the Rebuttal PDF** to compare the editing speed and performance of UniKE with baselines. It shows that UniKE achieves significant performance advantages without incurring additional resource costs or computation time.
> Therefore, we think that the additional complexity they bring to the editing process is acceptable.
>
> **(2.2)** Leveraging 15k additional triplets to train the encoders is a **one-time** pre-training process, taking less than an hour. Once pre-trained, the encoders are **frozen** for downstream knowledge editing tasks, **adding no training-time cost to knowledge editing.** As shown in **Table 10 in the Rebuttal PDF**, after removal of encoders without pre-training, the performance significantly decreases. So we think that the time cost of this one-time pre-training is acceptable. Furthermore, this pre-training is common in knowledge editing. Methods like MEND and SERAC also require pre-training, **with pre-training time often exceeding one day, which is 24 times longer than  UniKE pre-training.**
>
> ***[Please Refer to Table 10 in the Rebuttal PDF for the Results]***
>
> &nbsp;
>
> **Q3:** Did not find an experiment discussing /beta
>
> **A3:** We apologize for neglecting to discuss the value of $\beta$. In our experiments, we set $\beta$=1.0. And we conduct further experiments, ranging $\beta$ from 0.0 to 1.6 to evaluate the performance. As shown in **Figure 1 in the Rebuttal PDF**, within the range of 0.8 to 1.6, UniKE's performance shows no significant variation with changes in $\beta$. **This indicates that $\zeta$ is robust to the change of $\beta$**.
>
> ***[Please Refer to Figure 1 in the Rebuttal PDF for the Results]***
>
> &nbsp;
>
> **Q4:** Why would you add extra 10 kv pairs in the FFN of last 4 transformer layers in intrinsic knowledge editing? It seems that \zeta is quite sensitive.
>
> **A4:** Thank you for the question. We would like to address your concerns from two points.
>
> **(4.1)** The implementation detail for intrinsic knowledge editing is to add an extra 10 kv pairs to the FFN of the last four transformer layers. "extra" means that **we preserve the original parameters of the MLLM while achieving intrinsic knowledge editing through the additionally added kv pairs**. Moreover, We choose these hyperparameters because **we find them to be the most suitable** for multimodal editing. Of course, we maintain a comparable number of neurons and transformer layers in the T-Patcher baseline for fair comparison.
>
> **(4.2)** Figure 4(c) in our paper highlights the importance of selecting an appropriate method for constructing the editing direction $\zeta$. Specifically, using a random vector for $\zeta$ degrades performance due to the introduction of noise during editing, which hinders the MLLM from learning effectively. In contrast, our proposed method for constructing $\zeta$ shows a noticeable performance improvement (especially in generality), as we learn a generalizable editing direction from a large number of samples, addressing the inherent limitations of generality in intrinsic knowledge editing.
>
> Moreover, once completing the construction of $\zeta$, as seen from the above **A3**, adjusting its weight $\beta$ has little impact on editing performance.
> It indicates that **the constructed $\zeta$ itself is robust to weight variations, although its construction method is sensitive.**
>
> &nbsp;
>
> Once again, we express our heartfelt gratitude for your valuable suggestions! In the revised version of the paper, we will improve the descriptions to make them clearer.

---

> > ### Comment · Reviewer_4G2a · 2024-08-09
> >
> > The rebuttal have addressed most of my concerns. I will raise my score to 6.

---

> > > ### Author Response · Authors · 2024-08-10
> > >
> > > Thank you for raising the score. Your valuable suggestions greatly contribute to the quality of our paper.  And we are deeply grateful for your insightful suggestions on our work!

---

> ### Author Response · Authors · 2024-08-07
> **Tables 7, 10 and Figure 1 in the Rebuttal PDF**
>
> To facilitate your reading, we also paste our additional experimental results here, **which are consistent with the tables / figures in the rebuttal PDF**.
>
> &nbsp;
>
> **Table 7:** The computational speed, resource utilization and performance of each method. We use the average results of five metrics (Reliability, T-Generality, M-Generality, T-Locality, and M-Locality) as the performance measure.
> | Method    | GPU memory | Editing time for each sample | Avg performance |
> |:----------|:----------:|:----------------------------:|:---------------:|
> | FT        | 22G        | 6.1s                         | 60.6            |
> | KE        | 24G        | 5.8s                         | 74.7            |
> | T-Patcher | 18G        | 4.7s                         | 80.4            |
> | MEND      | 36G        | 5.2s                         | 90.3            |
> | IKE       | 20G        | 1.6s                         | 65.5            |
> | SERAC     | 49G        | 3.6s                         | 76.4            |
> | **UniKE**     | 18G        | 5.0s                         | **95.2**        |
>
> &nbsp;
>
> **Table 10:** Editing time cost and performance with/without encoders for UniKE. The time refers to the average editing or inference time for one sample. Gen is the average result of T-Generality and M-Generality; while Loc is the average result of T-Locality and M-Locality.
>
> | Method | GPU Memory | Editing time | Inference time | Rel. | Gen. | Loc. |
> | :--- | :---: | :---: | :---: | :---: | :---: | :---: |
> | w/o encoders | 17.7GB | 4.92s | 0.212s | 96.2 | 91.2 | 90.3 |
> | UniKE | 17.8GB | 5.04s | 0.217s | **97.4** | **94.6** | **93.5** |
>
> &nbsp;
>
> **Table corresponding to Figure 1 in Rebuttal PDF:**  Editing Performance on Different Value of $\beta$. Generality is the average result of T-Generality and M-Generality; while Locality is the average result of T-Locality and M-Locality.
> | $\beta$ | 0.0 | 0.4 | 0.8 | 1.0 | 1.2 | 1.6 |
> | :--- | :---: | :---: | :---: | :---: | :---: | :---: |
> | Rel. | 97.0 | 97.7 | 97.8 | 98.0 | 97.5 | 97.9 |
> | Gen. | 91.7 | 93.3 | 95.0 | 95.1 | 94.8 | 94.7 |
> | Loc. | 91.6 | 93.0 | 93.6 | 93.8 | 93.2 | 93.5 |
>
> &nbsp;
>
> We hope we have effectively addressed your concerns. Discussions are always open. Thank you once again for your time and insightful comments!

---

### Official Review · Reviewer_V1xr · 2024-07-07

**Soundness:** 4
**Presentation:** 4
**Contribution:** 4
**Rating:** 9
**Confidence:** 5

**Summary:**

This paper proposes UniKE, a novel multimodal editing method that establishes a unified perspective for intrinsic knowledge editing and external knowledge resorting. On this basis, the authors combine both types of knowledge editing methods, executing them in the latent space with a unified paradigm. Furthermore, this paper proposes to disentangle the knowledge representations into the semantic and truthfulness spaces, effectively enhancing the collaboration between intrinsic knowledge and external knowledge resorting. Extensive experimental results show that UniKE achieves promising results under various settings, ensuring that the post-edit MLLM maintains excellent reliability, generality, and locality.

**Strengths:**

(1) I think knowledge editing for MLLMs is a relatively new topic. Previously, a few studies merely adapted existing knowledge editing methods from the NLP field into multimodal domain. To the best of my knowledge, this paper is the first to conduct a detailed and systematic analysis of the strengths and weaknesses of existing methods when applied to editing multimodal LLMs.

(2) The proposed method is very novel and effective. Previous efforts in knowledge editing show significant differences between intrinsic knowledge editing methods and external knowledge resorting methods. In this work, the authors ingeniously convert in-context editing into the format of feature shifting, achieving a unification of the editing paradigms that can operate simultaneously within the same transformer layer with synergistic correlation. I find this to be a very inspiring design. Moreover, the design of knowledge collaboration is closely integrated with this unified paradigm.

(3) The experiments are very solid and thorough, clearly demonstrating that UniKE effectively addresses multimodal knowledge editing tasks under various setups. Meanwhile, the authors have also provided the implementation code for the experiments.

(4) I commend the authors for conducting an extensive set of ablations and analyses, which are very helpful in understanding the impact of each component within UniKE.

(5) Additionally, I believe that the method proposed by the authors is not only applicable to knowledge editing tasks. By converting in-context learning into the representation space and avoiding the need to increase the context window space, it better synergizes with parameter update learning. I consider this to have significant implications for further studies on how to construct more powerful MLLMs.

**Weaknesses:**

(1) In the NLP community, some studies will discuss the resilience to overediting [1] of knowledge editing methods by adopting the contrastive knowledge assessment [2] (CKA). Unlike the locality property that measures whether LLMs forget previous knowledge, overediting can be understood as excessive generalized to seemingly similar but unrelated samples. Although there may be no current work on multimodal editing that discusses the phenomenon of overediting, I encourage authors to add relevant experiments for a straightforward comparison of the resilience to overediting among each method (UniKE, MEND, T-Patcher, and IKE).
(2) A more challenging task of knowledge editing is counterfactual editing, where the edited answer $y$ to the question $x$ can sometimes be counterfactual to the real world. A typical counterfactual editing dataset in the NLP community is called COUNTERFACT [3], which more accurately reflects the true effectiveness of knowledge editing methods by avoiding the effects of LLMs knowing this knowledge before editing. I encourage authors to construct multimodal counterfactual editing datasets and conduct more experiments to verify whether UniKE performs better in counterfactual editing scenarios compared to MEND, T-Patcher and IKE.
[1] Zheng, Ce, et al. "Can we edit factual knowledge by in-context learning?." arXiv preprint arXiv:2305.12740 (2023).
[2] Dong, Qingxiu, et al. "Calibrating factual knowledge in pretrained language models." arXiv preprint arXiv:2210.03329 (2022).
[3] Meng, Kevin, et al. "Locating and editing factual associations in GPT." Advances in Neural Information Processing Systems 35 (2022): 17359-17372.

**Questions:**

See the weaknesses.

**Limitations:**

The authors have adequately discussed the limitations and potential negative societal impact.

---

> ### Author Rebuttal · Authors · 2024-08-07
>
> We sincerely thank you for your valuable comments and high appreciation of our work! We are encouraged that our research is recognized as having significant implications for further studies on constructing more powerful MLLMs. We will address your concerns point by point.
>
> **Q1:** In the NLP community, some studies will discuss the resilience to overediting of knowledge editing methods by adopting the contrastive knowledge assessment (CKA). Unlike the locality property that measures whether LLMs forget previous knowledge, overediting can be understood as excessively generalized to seemingly similar but unrelated samples. Although there may be no current work on multimodal editing that discusses the phenomenon of overediting, I encourage authors to add relevant experiments for a straightforward comparison of the resilience to overediting among each method (UniKE, MEND, T-Patcher, and IKE).
>
> **A1:** Thank you for raising such a professional question! To construct the dataset for overediting evaluation, we randomly select a portion of the test data from the E-VQA task. We then make minor modifications to the text questions in each example so that they appear similar to the original questions but differ in semantics, thus constructing the test data for overediting. To improve experimental efficiency, we conduct experiments on MiniGPT-4 with the setup of one-step editing. The results of the CKA evaluation for each method are listed in the following table. We can see that UniKE outperforms all other methods. The results indicate that **UniKE has less influence on over-editing, which further demonstrates its robust capabilities.
>
> Table 1: The results of CKA evaluation on MiniGPT-4 with the setup of one-step editing.
> | Method | T-Patcher | MEND | IKE |  UniKE |
> |:-|:-:|:-:|:-:|:-:
> | **CKA** | 1.38 | 1.33| 1.47  | **1.58** |
>
> We also present the results in Table 8 of the Rebuttal PDF.
>
> &nbsp;
>
> **Q2:** A more challenging task of knowledge editing is counterfactual editing, where the edited answer to the question can sometimes be counterfactual to the real world. A typical counterfactual editing dataset in the NLP community is called COUNTERFACT, which more accurately reflects the true effectiveness of knowledge editing methods by avoiding the effects of LLMs knowing this knowledge before editing. I encourage authors to construct multimodal counterfactual editing datasets and conduct more experiments to verify whether UniKE performs better in counterfactual editing scenarios compared to MEND, T-Patcher, and IKE.
>
> **A2:** Thank you for the insightful question. We first construct a counterfactual editing dataset based on the MMEdit dataset, ensuring that the MLLM did not have prior knowledge of the editing target. We also conduct experiments on MiniGPT-4 with one-step editing. The results of the counterfactual editing are shown in the following table. **It can be seen that UniKE significantly outperforms existing methods in this more challenging editing scenario, fully demonstrating the effectiveness of our method.
>
> Table 2: Performance of counterfactual editing on MiniGPT-4.
> | Method | Rel | T-Gen | M-Gen | T-Loc | M-Loc | Avg |
> | :-   | :-:|:-:|:-:|:-:|:-:|:-:|
> | T-Patcher|  80.0 | 65.9 | 57.7 | 84.2 | 88.3 | 75.2 |
> | MEND | 90.6 | 83.2 | 74.1 | 93.5 | 82.1 | 84.7 |
> | IKE | 90.3 | 83.7 | **81.5** | 44.1 | 5.0 | 60.9 |
> | UniKE  | **90.8** | **84.7** | 80.7 | **94.9** | **94.5** | **89.1** |
>
> We also present the results in Table 9 of the Rebuttal PDF.
>
> &nbsp;
>
> We will add these experiments to the main body or the appendix of our paper. Thank you once again for your insightful and professional feedback!

---

> ### Comment · Reviewer_V1xr · 2024-08-10
>
> Thank you for your rebuttal response. I think that over-editing evaluation and counterfactual editing are critical tasks that can reflect whether a model truly possesses the capabilities of knowledge cognitive learning in knowledge editing scenarios. It's impressive to see that UniKE performs well on these two challenging tasks. I consider UniKE to be a strong contribution, and I will raise my score to 9.

---

> > ### Author Response · Authors · 2024-08-10
> >
> > Thank you for raising the score. We deeply appreciate your recognition of our work and the constructive advice you've offered!

---

### Official Review · Reviewer_w8aR · 2024-07-07

**Soundness:** 3
**Presentation:** 3
**Contribution:** 3
**Rating:** 6
**Confidence:** 3

**Summary:**

UniKE is a unified framework for multi-modal knowledge editing that includes three main aspects:
1.Knowledge Separation: UniKE divides knowledge into factuality and semantic spaces to manage and coordinate different types of knowledge more effectively.
2.Knowledge Collaboration: In the factuality space, UniKE standardizes new knowledge based on a learned factuality distribution, enhancing reliability and generality. In the semantic space, it adjusts the integration of external knowledge based on relevance to the input samples, maintaining locality.
3.Multi-step Editing: UniKE supports single-step, multi-step sequence, and cross-task editing while maintaining high reliability, generality, and locality.
Through these innovations, UniKE significantly improves performance in multi-modal knowledge editing tasks.

**Strengths:**

1.The paper introduces UniKE, a novel framework that seamlessly integrates intrinsic and external knowledge editing, enhancing the model’s ability to handle complex multimodal information effectively. The motivation behind the work is clear, and the article is well structured, guiding the reader through the innovative approach and its benefits.

2.By disentangling knowledge into semantic and truthfulness spaces, the proposed method ensures robust collaboration between different types of knowledge, significantly improving the model's reliability, generality, and locality. The method's effectiveness is demonstrated through comprehensive experiments across various settings, consistently outperforming existing state-of-the-art methods.

3.UniKE's design allows for application across different multimodal models and editing scenarios, making it a versatile and robust solution for enhancing multimodal language models.

**Weaknesses:**

1.The paper does compare UniKE with other intrinsic knowledge editing and external knowledge resorting methods and highlights its efficiency in several aspects. However, it lacks a detailed discussion on computational speed and resource utilization.

2.The reliance on fine-tuning for knowledge updates could lead to overfitting, especially if the model is frequently updated. This could impact the model’s generalization abilities, making it less effective in unforeseen or less frequent scenarios.

3.The paper lacks detailed explanations for the evaluation metrics in Line 240. In Table 2, across multiple experiments, it is unclear why there is little difference compared to the SERAC method in the T-Loc metric, but a significant difference in the M-Loc metric.

4.The paper lacks detailed information about the parameter "n" used in the contrastive learning formula, making it difficult to understand its impact on model performance, and does not discuss how different values of "n" might influence the results and effectiveness of the method.

**Questions:**

Please address the comments in the weaknesses section.

**Limitations:**

The limitations of the article have been discussed by the authors

---

> ### Author Rebuttal · Authors · 2024-07-31
>
> Thank you for your kind feedback and valuable comments. We will explain your concern as follows.
>
> **Q1:** Lacking a detailed discussion on computational speed and resource utilization.
>
> **A1:** Thank you for the suggestion. In **Table 7 of the Rebuttal PDF**, we list the computational speed and resource utilization of each method. As can be seen, compared to most baselines, **UniKE achieves significant performance advantages without incurring additional resource costs or computation time**.  Moreover, although external knowledge editing methods (IKE, SERAC) are faster in terms of editing speed, their performance is unacceptable, **lagging 20-30 points behind UniKE**. These results demonstrate the superiority of UniKE.
>
> ***[Please Refer to Table 7 in the Rebuttal PDF for the Results]***
>
> &nbsp;
>
> **Q2:** The reliance on fine-tuning for knowledge updates could lead to overfitting.
>
> **A2:** First, The objective and evaluation metrics of knowledge editing include not only successfully completing edits but also ensuring the post-edit MLLM can generalize over equivalent input neighbors (**Generality**) and maintain consistent output for irrelevant inputs (**Locality**). In fact, **Generality and Locality can be considered as two aspects of evaluating overfitting: the new knowledge should generalize well without disrupting the generalization of unrelated old knowledge.**
>
> In our method, to maintain Generality and Locality and prevent overfitting, **we retain the original model parameters while combining both intrinsic and external knowledge editing within a unified framework**. Intrinsic knowledge editing **introduces a minimal amount of tunable parameters** in a small subset of FFNs **to preserve locality**. Meanwhile, external knowledge editing incorporates retrieved in-context representations (with no tunable parameters) into the latent space of the transformer **to enhance generality**.  We also propose knowledge disentangling to promote knowledge collaboration.
> In the semantic space, intrinsic knowledge helps select appropriate external knowledge, **preventing locality disruption**. In the truthfulness space, external knowledge identifies a generalizable editing direction, regulating intrinsic knowledge and **alleviating its restriction on generality**.
>
> The experimental results (Tables 2, 3, and 4 in our paper) demonstrate that UniKE achieves better locality and generality in various multimodal editing scenarios, **and more effectively addresses the issue of overfitting compared to baselines.**
>
> &nbsp;
>
> **Q3:** The paper lacks detailed explanations for the evaluation metrics. In Table 2, across multiple experiments, it is unclear why there is little difference compared to the SERAC method in the T-Loc metric, but a significant difference in the M-Loc metric.
>
> **A3:** We apologize for any confusion regarding the evaluation metrics. In Appendix B.1, we detail the definitions of the five evaluation metrics (see Eq.11-15).
> In brief, these metrics assess accuracy across different input samples: Reliability on edit targets, T-Locality on unrelated QA tasks, M-Locality on unrelated VQA tasks, T-Generality on samples that rephrase the text inputs based on the edit targets, and M-Generality on samples that redraw image inputs based on the edit targets.
>
> Furthermore, To analyze the reasons for the little differences in the T-Loc metric and the significant differences in M-Loc, we examined some intermediate outputs of SERAC during editing. We discover that the reason is **related to the pipeline of SERAC**.
> SERAC adopts a counterfactual model while keeping the original model unchanged. It employs a scope classifier to determine if new input falls within the range of stored edit examples. If the input matches any cached edit, output the counterfactual model’s prediction based on the input and the most probable edit. Otherwise, the original model’s prediction is given.
>
> For T-Loc, the input samples contain only text information, which differs significantly from the input format of edited samples. This makes the counterfactual model less likely to activate, resulting in outputting the original model's prediction and better locality. However, for M-Loc, the input samples, like the edited samples, contain multimodal information. As the scope classifier is trained to activate the counterfactual model for multimodal inputs even if they are unrelated to the editing target, **the incorrect choices made by the scope classifier result in significantly worse M-Loc performance of SERAC, causing the observed phenomenon.**
>
> &nbsp;
>
> **Q4:** The paper lacks detailed information about the parameter "n" used in the contrastive learning formula, and does not discuss how different values of "n" might influence the results and effectiveness of the method.
>
> **A4:** Thank you for raising an important concern! "n" is a crucial hyperparameter worth discussing. First, n represents the number of training samples used for contrastive learning during encoder pre-training. These training data come from the knowledge representations of the in-context knowledge we constructed, which we detailed in Appendix C on how to prepare and extract these in-context knowledge representations.
> In our experiments, we set n to approximately 16000.
>
> Building on this, we perform an ablation study on the value of n to test how different numbers of pre-training samples affect the final editing performance. As shown in **Figure 2 in the Rebuttal PDF**, the performance keeps increasing when the data number increases from 0 to 8000. Beyond this, escalating the data count from 8000 to 16000  yields only marginal enhancement. **It demonstrates that our encoder training is data-efficient and only requires relatively small amounts of data to achieve effective knowledge representation disentangling.**
>
> ***[Please Refer to Figure 2 in the Rebuttal PDF for the Results]***
>
> &nbsp;
>
> Thank you once again for your time and valuable feedback!

---

> ### Author Response · Authors · 2024-08-01
>
> Dear Reviewer:
> ﻿
> Thank you very much for your kind feedback and valuable comments. I have carefully read your comments. However, it appears that the content from the "Strengths" section may have been inadvertently pasted into the "Weaknesses" section. As a result, the weaknesses you mentioned still seem to describe the strengths of our work. This makes it challenging to identify and address the specific concerns you may have intended to highlight. ﻿
> ﻿
> Could you please review your comments once more and provide additional clarification on the weaknesses you observed? Your insights are crucial for improving our paper, and I am keen to address all your concerns effectively. ﻿
> ﻿
> Thank you again for your time and valuable feedback!

---

> > ### Comment · Reviewer_w8aR · 2024-08-01
> > **Clarification on Weakness**
> >
> > I apologize for the mistake in my previous feedback. I have reviewed my comments and provided the correct feedback on the weaknesses of your paper. I hope this clarifies my observations and helps you improve your paper.

---

> ### Author Response · Authors · 2024-08-07
> **Table 7 and Figure 2 in the Rebuttal PDF**
>
> To facilitate your reading, we also paste our additional experimental results here, **which are consistent with the tables / figures in the rebuttal PDF**.
>
> &nbsp;
>
> **Table 7:** The computational speed, resource utilization and performance of each method. We use the average results of five metrics (Reliability, T-Generality, M-Generality, T-Locality, and M-Locality) as the performance measure.
> | Method    | GPU memory | Editing time for each sample | Avg performance |
> |:----------|:----------:|:----------------------------:|:---------------:|
> | FT        | 22G        | 6.1s                         | 60.6            |
> | KE        | 24G        | 5.8s                         | 74.7            |
> | T-Patcher | 18G        | 4.7s                         | 80.4            |
> | MEND      | 36G        | 5.2s                         | 90.3            |
> | IKE       | 20G        | 1.6s                         | 65.5            |
> | SERAC     | 49G        | 3.6s                         | 76.4            |
> | **UniKE**     | 18G        | 5.0s                         | **95.2**        |
>
> &nbsp;
>
> **Table corresponding to Figure 2 in Rebuttal PDF:** Editing performance on different value of $n$. Generality is the average result of T-Generality and M-Generality; while Locality is the average result of T-Locality and M-Locality.
> | $n$ | 0 | 4000 | 8000 | 12000 | 16000 |
> | :--- | :---: | :---: | :---: | :---: | :---: |
> | Rel. | 96.2 | 97.0 | 97.8 | 98.1 | 98.0 |
> | Gen. | 91.2 | 93.3 | 94.5 | 94.9 | 95.1 |
> | Loc. | 90.3 | 92.2 | 93.3 | 93.7 | 93.8 |
>
> &nbsp;
>
> We hope we have addressed all of your concerns. Discussions are always open.  Thank you once again for your constructive suggestions!

---

### Official Review · Reviewer_mp35 · 2024-07-13

**Soundness:** 3
**Presentation:** 3
**Contribution:** 3
**Rating:** 7
**Confidence:** 4

**Summary:**

This paper proposes UniKE, a novel multimodal editing method that establishes a unified perspective and paradigm for intrinsic knowledge editing and external knowledge resorting. Within such a unified framework, the authors further promote knowledge collaboration by disentangling the knowledge representations into the semantic and truthfulness spaces. Extensive experiments validate the effectiveness of UniKE, which ensures that the post-edit MLLM simultaneously maintains excellent reliability, generality, and locality.

**Strengths:**

1. The paper is well-written and easy to follow and the motivation is clear and reasonable.
2. The paper gives a unified perspective on the intrinsic refinement of knowledge and the strategic reorganization of external knowledge, which can enhance subsequent research endeavors.
3. The experimental settings (one-step editing, sequential editing, cross-task editing) are fair.

**Weaknesses:**

1. The authors only edit Qformer style MLLMs (MiniGPT4, BLIP2), more foundation models should be compared. e.g., LLava. Besides, the improvement on BLIP2 (Tab.1) seems incremental.
2. More recent model editing baselines should be compared to prove the effectiveness of the proposed methods.
3. The paper focuses on MLLM editing, what are the main differences between MLLM editing and LLM editing? What are the specific designs for multimodal models? I notice that some compared methods are proposed for LLM editing. Can the proposed methods be used on LLM editing? The authors should give some explanations and experimental results if possible.

**Questions:**

Please refer to the weakness below.

**Limitations:**

The paper briefly mentioned limitations.

---

> ### Author Rebuttal · Authors · 2024-08-07
>
> We sincerely thank you for the valuable comments! We are encouraged to see that our work can enhance subsequent research endeavors. We will address your concerns point by point.
>
> **Q1**: More foundation models should be compared (LLava). The improvement on BLIP2 seems incremental.
>
> **A1**: Thank you for raising the important concern. We first leverage LLaVA1.5 to perform multimodal editing, and then we will show that the improvement on BLIP-2 OPT is not incremental.
>
> **(1.1)Multimodal Editing on LLaVA**
>
> We conduct both one-step editing and 10-step cross-task editing on LLaVA1.5. The results are shown in **Tables 1 and 2 of the rebuttal PDF**. With LLaVA as the backbone, **UniKE still balances all three target properties well, outperforming all baseline methods** in both one-step editing and cross-task editing. This demonstrates that UniKE is **model-agnostic and effective across various types of MLLMs**.
>
> ***[Please Refer to Tables 1 and 2 in the Rebuttal PDF for the Results]***
>
> &nbsp;
>
> **(1.2)Analysis on BLIP-2 OPT**
>
> Though improvement of one-step editing on Blip-2 OPT(in Table1 of our paper) may seem incremental, we aim to demonstrate the significant performance advantage of UniKE on BLIP-2 OPT from 3 aspects:
>
> (I) Across all ten metrics of BLIP-2 OPT one-step editing, UniKE achieves best results in five metrics, with the remaining suboptimal results only marginally lower than the best. We also calculate the average performance of each method across all ten metrics, as shown in the following Table. **The average result of UniKE in BLIP-2 OPT one-step editing significantly surpasses other methods, exceeding the second-best method (MEND) by 5.1 points**.
>
> Table: Avg performance of all metrics on Blip-2 OPT one-step editing
> |Method|FT|KE|TPatcher|MEND|IKE|SERAC|**UniKE**
> |:-|:-:|:-:|:-:|:-:|:-:|:-:|:-:
> |AVG Perf.|56.2|70.3|82.3|90.5|65.3|77.6|**95.6**|
>
> (II) In one-step editing on BLIP-2 OPT, **UniKE is the only method that consistently performs well across all metrics, whereas other methods have at least one notably poor metric**. For instance, MEND's M-Gen(E-VQA), M-Gen(E-IC), and M-Loc(E-IC) results are 15.7, 19.0, and 14.9 points lower than UniKE's, respectively. IKE and SERAC perform especially poorly on M-Loc, with accuracy below 10%.
>
> (III) Additionally, the relatively low difficulty of one-step editing task may allow some baselines to maintain relatively high evaluation scores. In more challenging scenarios such as multi-step cross-task editing, UniKE's advantage becomes more apparent. Specifically, in **BLIP-2 OPT cross-task editing** with higher difficulty (shown in  **Table 3 of the Rebuttal PDF**), **UniKE consistently outperforms all other baselines in all metrics, demonstrating a more significant performance advantage**.
>
> ***[Please Refer to Table 3 in the Rebuttal PDF for the Results]***
>
> &nbsp;
>
> **Q2**: More recent baselines should be compared.
>
> **A2**: Thank you for the suggestion. We further compare UniKE with two recent baselines proposed in 2024: MENMET, WISE (a concurrent work). We leverage MiniGPT4 to conduct both one-step editing and cross-task editing. The results are shown in **Tables 4 and 5 of the Rebuttal PDF**. **The performance of UniKE surpasses these recent baselines, further demonstrating the effectiveness of UniKE.**
>
> [Tan et al 2024]MENMET: Massive Editing for Large Language Models via Meta Learning.ICLR2024.
>
> [Wang et al 2024]WISE: Rethinking the Knowledge Memory for Lifelong Model Editing of Large Language Models.23 May 2024.
>
> ***[Please Refer to Tables 4 and 5 in the Rebuttal PDF for the Results]***
>
> &nbsp;
>
> **Q3**: What are the main differences between MLLM editing and LLM editing?What are the specific designs for multimodal models?Can UniKE be used on LLM editing?
>
> **A3**: Thank you for your questions. We will address them from three points:
>
> **(3.1)** The main differences between MLLM editing and LLM editing lie in **task difficulty**. LLM stores single-modality NLP knowledge within its parameters, while MLLM stores multi-modality knowledge, making MLLM editing more challenging due to the need to edit knowledge from multiple modalities to fix errors. **Although mainstream editing methods proposed for both are similar, current methods can effectively edit LLMs but not MLLMs** [Chen et al 2023].
>
> **(3.2)** [Chen et al 2023] attempted specific designs for editing MLLMs such as **editing the unique Qformer**, but found **these methods less effective than directly applying LLM editing methods**, as shown in the following table. This is because the bottleneck for MLLMs lies in enabling LLMs to reason with multimodal input, rather than in extracting visual information. Therefore, **mainstream MLLM editing methods still focus on editing LLM and directly adopt LLM editing techniques.**.
>
> However, directly applying LLM editing methods for multimodal editing still fails to balance both generality and locality. In this paper, we discuss the shortcomings of existing LLM editing methods when applied to multimodal scenarios and develop a unified framework along with knowledge disentangling, which leverages the strength and mitigates the weakness of each method type, leading to more effective multimodal editing.
>
> ||Methods editing Qformer|Methods editing LLM|UniKE
> |:-|:-:|:-:|:-:
> |AVG Multimodal Editing Performance|71.8|89.3|**95.2**|
>
> **(3.3)** Since LLM editing is merely a simpler case of MLLM editing, UniKE which successfully addresses MLLM editing, can also be effectively used for LLM editing. Following previous LLM-editing work, we use GPT-J-6B to conduct both one-step and multi-step editing on the ZsRE benchmark. As shown in **Table 6 of the Rebuttal PDF, UniKE still outperforms all baselines in LLM editing**.
>
> ***[Please Refer to Table 6 in the Rebuttal PDF for the Results]***
>
> &nbsp;
>
> We will integrate these experiments into our paper. Thank you again for the valuable feedback!
>
> [Chen et al 2023]Can We Edit Multimodal Large Language Models?EMNLP2023.

---

> > ### Comment · Reviewer_mp35 · 2024-08-09
> > **Official Comment**
> >
> > Thanks for your response. The Rebuttal addresses most of my concerns. I will raise my score to 7.

---

> > > ### Author Response · Authors · 2024-08-09
> > >
> > > Thank you for raising the score. Your valuable suggestions greatly contribute to the quality of our manuscript. Thank you again for your precious time and valuable suggestions!

---

> ### Author Response · Authors · 2024-08-07
> **Tables 1-6 in the Rebuttal PDF**
>
> To facilitate your reading, we also paste our additional experimental results here, **which are consistent with the tables in the rebuttal PDF**.
>
> &nbsp;
>
> **Table 1**: Performance of one-step editing on LLaVA1.5 (We average the results on E-IC and E-VQA).
>
> | Method | Rel. | T-Gen. | M-Gen. | T-Loc. | M-Loc. | **Avg** |
> | :--- | :---: | :---: | :---: | :---: | :---: | :---: |
> | FT | 67.4 | 57.9 | 54.8 | 67.1 | 63.2 | 62.1 |
> | KE | 72.7 | 65.4 | 55.3 | 82.2 | 67.3 | 68.4 |
> | T-Patcher | 89.0 | 76.5 | 69.0 | 81.2 | 81.3 | 79.6 |
> | MEND | 95.4 | 92.6 | 78.3 | 83.5 | 80.3 | 86.0 |
> | IKE | 93.4 | 85.1 | 77.9 | 27.7 | 3.2 | 57.5 |
> | SERAC | **96.3** | 92.4 | 85.5 | 83.3 | 7.7 | 73.0 |
> | **UniKE** | 95.7 | **92.8** | **88.4** | **86.0** | **86.4** | **89.9** |
>
> &nbsp;
>
> **Table 2**: Performance of cross-task editing on LLaVA1.5.
>
> | Method | Rel. | T-Gen. | M-Gen. | T-Loc. | M-Loc. | Avg |
> | :--- | :---: | :---: | :---: | :---: | :---: | :---: |
> | FT | 66.9 | 57.2 | 51.0 | 62.3 | 54.4 | 56.4 |
> | KE | 69.8 | 60.3 | 52.5 | 79.3 | 62.1 | 64.8 |
> | T-Patcher | 81.2 | 60.0 | 57.4 | 77.4 | 76.5 | 70.5 |
> | MEND | 90.4 | 84.3 | 73.8 | 78.6 | 76.0 | 80.6 |
> | SERAC | 92.1 | 88.3 | 82.5 | 82.2 | 1.2 | 69.3 |
> | **UniKE** | **92.2** | **89.2** | **83.8** | **82.7** | **84.7** | **86.5** |
>
> &nbsp;
>
> **Table 3**: Performance of cross-task editing on BLIP-2 OPT.
> | Method | Rel. | T-Gen. | M-Gen. | T-Loc. | M-Loc. | Avg |
> | :--- | :---: | :---: | :---: | :---: | :---: | :---: |
> | FT | 57.2 | 49.9 | 43.2 | 52.2 | 49.7 | 50.4 |
> | KE | 64.2 | 60.1 | 57.2 | 83.5 | 59.2 | 64.8 |
> | T-Patcher | 83.1 | 69.7 | 65.9 | 84.5 | 77.9 | 76.2 |
> | MEND | 84.2 | 82.4 | 74.9 | 91.4 | 80.2 | 82.6 |
> | SERAC | 90.8 | 89.2 | 84.1 | 90.0 | 1.7 | 71.2 |
> | **UniKE** | **91.1** | **90.6** | **88.2** | **91.7** | **85.6** | **89.4** |
>
> &nbsp;
>
> **Table 4**: Comparison with recent baselines for one-step editing on MiniGPT-4 (We average the results on E-IC and E-VQA).
> | Method | Rel. | T-Gen. | M-Gen. | T-Loc. | M-Loc. | Avg |
> | :--- | :---: | :---: | :---: | :---: | :---: | :---: |
> | MENMET | 97.0 | 96.2 | 82.4 | 98.0 | 85.2 | 91.8 |
> | WISE | 97.2 | 92.2 | 88.7 | 98.4 | **88.2** | 93.0 |
> | **UniKE** | **97.4** | **96.6** | **92.6** | **98.8** | 88.1 | **94.7** |
>
> &nbsp;
>
> **Table 5**: Comparison with recent baselines for cross-task editing on MiniGPT-4.
> | Method | Rel. | T-Gen. | M-Gen. | T-Loc. | M-Loc. | **Avg** |
> | :--- | :---: | :---: | :---: | :---: | :---: | :---: |
> | MENMET | 88.4 | 87.2 | 78.0 | 86.1 | 82.5 | 84.4 |
> | WISE | 89.2 | 85.4 | 83.4 | 87.8 | 83.6 | 85.9 |
> | **UniKE** | **90.7** | **88.2** | **86.8** | **90.4** | **83.8** | **88.0** |
>
> &nbsp;
>
> **Table 6**: Performance of each method on LLM editing task (ZsRE) for one-step editing and 200-step editing.
> |  | ONE-STEP EDITING | | ||200-STEP EDITING | | | |
> | :---   | :---:| :---: | :---:  | :---: | :---: |  :---: | :---: |  :---: |
> | **Method** |**Rel.** | **Gen.** | **Loc.** | **Avg.** | **Rel.** | **Gen.** | **Loc.** | **Avg.** |
> | FT  | 77.4 | 76.7 | 35.5 | 63.2 | 19.5 | 17.2 | 5.4 | 14.0
> | KE  | 20.6 | 20.1 | 81.3 | 40.7 | 7.6 | 6.8 | 65.8 |  26.7
> | T-Patcher  | 97.1 | 95.0 | 96.2 | 96.1 | 81.4 | 70.6 | 91.3 | 81.1
> | MEND  | 98.2 | 97.7 | 97.4 | 97.8 | 0.0 | 0.0 | 0.0 | 0.0 |
> | In-Context Editing  | 99.4 | 97.2 | 59.2 | 85.3 | - | - | - | - |
> | SERAC  | 88.6  | 87.9 | 99.9 | 92.1 | 24.0 | 23.2 | **96.4** | 47.9
> | MENMET  | 99.1 | 86.8 | 97.4 | 94.4 | 82.9 | 73.6 | 90.2 | 82.2
> | WISE  | 98.8 | 96.3 | **99.9** | 98.3 | 82.8 | 74.7 | 95.5 | 84.3
> | **UniKE**  | **99.5** | **97.9** | 99.6 | **99.0** | **85.1** |  **76.7** | 95.6 | **85.8** |
>
> &nbsp;
>
> These experiments will be integrated into the main body or the appendix of our paper. We hope we have addressed all of your concerns. Discussions are always open. Thank you again for your time and valuable suggestions!

---

### Author Rebuttal · Authors · 2024-08-07

We sincerely thank all the reviewers for their insightful and valuable comments! Overall, we are encouraged that they find that:

(1) The motivation is **clear and reasonable**, supported by a well-structured article. *(Reviewer mp35, Reviewer w8aR, Reviewer V1xr)*

(2) UniKE establishes **a unified framework** for intrinsic knowledge editing and external knowledge resorting, which is **novel and effective**. *(All Reviewers)*

(3) The experiments are **solid and thorough**, clearly demonstrating that UniKE consistently maintains excellent reliability, generality, and locality **across various settings**. *(All Reviewers)*

(4) UniKE has **significant implications for further studies** on constructing more powerful MLLMs, can **enhance subsequent research endeavors**, and is a **versatile and robust solution** for enhancing multimodal language models. *(Reviewer mp35, Reviewer w8aR, Reviewer V1xr)*

&nbsp;

To address the concerns raised by the reviewers, we have conducted several additional experiments to further demonstrate the superiority of UniKE from various perspectives. And we include these experimental results **in the rebuttal PDF**, which contains 10 tables and 2 figures.

(1) In Tables 1-3, we perform one-step editing and cross-task editing with LLaVA 1.5 as the backbone, also conducting cross-task editing on Blip-2 OPT. The results demonstrate that UniKE is effective across various types of MLLMs.

(2) In Tables 4-5, we compare UniKE with more recent baselines, further demonstrating the proficiency of our proposed UniKE.

(3) In Table 6, we apply UniKE to LLM editing tasks, showing that UniKE can effectively address LLM editing tasks in pure NLP scenarios.

(4) In Tables 8-9, we leverage UniKE on two interesting and challenging tasks with newly constructed data (overediting evaluation and counterfactual editing), finding that UniKE minimizes the influence of overediting and effectively addresses counterfactual editing tasks.

(5) In Table 7, Table 10, and Figures 1-2, we further analyze the editing time efficiency of UniKE and provide additional ablation studies, proving the robustness of UniKE's design.

&nbsp;

These experiments will be integrated into the main body or the appendix of our paper. Next, we will address each reviewer's detailed concerns point by point. We hope to address all of your concerns. Discussions are always welcome. Thank you!

---

### Decision · Program_Chairs · 2024-09-25

**Decision:**

Accept (spotlight)

**Comment:**

All reviewers acknowledged the novelty of this paper, as well as the solid and thorough experiments conducted. In the initial reviews, there were suggestions to further demonstrate the superiority of the proposed method from various perspectives. After rebuttal phase, the paper received positive feedback from all reviewers. AC has also carefully reviewed the paper, the reviews, and the rebuttal, and concurs with the reviewers' opinions. Therefore, AC recommends acceptance. It would be greatly beneficial to include the additional results from the rebuttal phase in the final version.